# Decodable and Sample Invariant Continuous Object Encoder

**Dehao Yuan, Furong Huang, Cornelia Fermüller & Yiannis Aloimonos**
Department of Computer Science
University of Maryland
College Park, MD 20740, USA
{dhyuan, furongh, fermulcm, jyaloimo}@umd.edu

## Abstract

We propose Hyper-Dimensional Function Encoding (HDFE). Given samples of a continuous object (e.g. a function), HDFE produces an explicit vector representation of the given object, invariant to the sample distribution and density. Sample distribution and density invariance enables HDFE to consistently encode continuous objects regardless of their sampling, and therefore allows neural networks to receive continuous objects as inputs for machine learning tasks, such as classification and regression. Besides, HDFE does not require any training and is proved to map the object into an organized embedding space, which facilitates the training of the downstream tasks. In addition, the encoding is decodable, which enables neural networks to regress continuous objects by regressing their encodings. Therefore, HDFE serves as an interface for processing continuous objects.

We apply HDFE to function-to-function mapping, where vanilla HDFE achieves competitive performance with the state-of-the-art algorithm. We apply HDFE to point cloud surface normal estimation, where a simple replacement from PointNet to HDFE leads to 12% and 15% error reductions in two benchmarks. In addition, by integrating HDFE into the PointNet-based SOTA network, we improve the SOTA baseline by 2.5% and 1.7% on the same benchmarks.

## 1 Introduction

Continuous objects are objects that can be sampled with arbitrary distribution and density. Examples include point clouds (Guo et al., 2020), event-based vision data (Gallego et al., 2020), and sparse meteorological data (Lu et al., 2021). A crucial characteristic of continuous objects, which poses a challenge for learning, is that their sample distribution and size varies between training and test sets. For example, point cloud data in the testing phase may be sparser or denser than that in the training phase. A framework that handles this inconsistency is essential for continuous object learning.

When designing the framework, four properties are desirable: (1) *Sample distribution invariance*: the framework is not affected by the distribution from which the samples are collected. (2) *Sample size invariance*: the framework is not affected by the number of samples. (3) *Explicit representation*: the framework generates outputs with fixed dimensions, such as fixed-length vectors. (4) *Decodability*: the continuous object can be reconstructed at arbitrary resolution from the representation.

Sample invariance (properties 1 and 2) ensures that differently sampled instances of the same continuous objects are treated consistently, thereby eliminating the ambiguity caused by variations in sampling. An explicit representation (property 3) enables a neural network to receive continuous objects as inputs, by consuming the encodings of the objects. Decodability (property 4) enables a neural network to predict a continuous object, by first predicting the representation and then decoding it back to the continuous object. Fig. 1 illustrates the properties and their motivations.

However, existing methodologies, which we divide into three categories, are limited when incorporating the four properties. (1) *Discrete framework*. The methods discretize continuous objects and process them with neural networks. For example, Liu et al. (2019) uses a 3D-CNN to process voxelized point clouds, Kim et al. (2017) uses an RNN to predict particle trajectories. These methods are not sample invariant – the spatial and temporal resolution must be consistent across the training

Figure 1: **Left**: HDFE encodes continuous objects into fixed-length vectors without any training. The encoding is not affected by the distribution and size with which the object is sampled. The encoding can be decoded to reconstruct the continuous object. **Right**: Applications of HDFE. HDFE can be used to perform machine learning tasks (e.g. classification, regression) on continuous objects. HDFE also enables neural networks to regress continuous objects by predicting their encodings.

and testing phases. (2) *Mesh-grid-based framework*. They operate on continuous objects defined on mesh grids and achieve discretization invariance (the framework is not affected by the resolution of the grids). Examples include the Fourier transform (Salih, 2012) and the neural operator (Li et al., 2020a). But they do not apply to sparse data like point clouds. (3) *Sparse framework*. They operate on sparse samples drawn from the continuous object. Kernel methods (Hofmann et al., 2008) work for non-linear regression, classification, etc. But they do not provide an explicit representation of the function. PointNet (Qi et al., 2017a) receives sparse point cloud input and produces an explicit representation, but the representation is not decodable (see Appendix B). In addition, all the frameworks require extra training of the encoder, which is undesired in some scarce data scenarios.

Currently, only the vector function architecture (VFA) (Frady et al., 2021) can encode an explicit function into a vector through sparse samples, while preserving all four properties. However, VFA is limited by its strong assumption of the functional form. VFA requires the input function to conform to $f(x) = \sum_k \alpha_k \cdot K(x, x_k)$, where $K : X \times X \to \mathbb{R}$ is a kernel defined on $X$. If the input function does not conform to the form, VFA cannot apply or induces large errors. In practice, such requirement is rarely satisfied. For example, $f(x)$ cannot even approximate a constant function $g(x) = 1$: to approximate the constant function, the kernel $K$ must be constant. But with the constant kernel, $f(x)$ cannot approximate other non-constant functions. Such limitation greatly hinders the application of VFA. Kindly refer to Appendix C for failure cases and detailed discussions.

We propose hyper-dimensional function encoding (HDFE), which does not assume any explicit form of input functions but only requires Lipschitz continuity (Appendix D illustrates some suitable input types). Consequently, HDFE can encode *a much larger class of functions*, while holding all four properties without any training. Thanks to the relaxation, HDFE can be applied to multiple real-world applications that VFA fails, which will be elaborated on in the experiment section. HDFE maps the samples to a high-dimensional space and computes weighted averages of the samples in that space to capture collective information of all the samples. A challenge in HDFE design is maintaining sample invariance, for which we propose a novel iterative refinement process to decide the weight of each sample. The contributions of our paper can be summarized as follows:

- We present HDFE, an encoder for continuous objects without any training that exhibits sample invariance, decodability, and distance-preservation. To the best of our knowledge, HDFE is the only algorithm that can encode Lipschitz functions while upholding all the four properties.
- We provide extensive theoretical foundation for HDFE. We prove that HDFE is equipped with all the desirable properties. We also verify them with empirical experiments.
- We evaluate HDFE on mesh-grid data and sparse data. In the mesh-grid data domain, HDFE achieves competitive performance as the specialized state-of-the-art (SOTA) in function-to-function mapping tasks. In the sparse data domain, replacing PointNet with HDFE leads to average error decreases of 12% and 15% in two benchmarks, and incorporating HDFE into the PointNet-based SOTA architecture leads to average error decreases of 2.5% and 1.7%.

## 2 PROBLEM DEFINITION AND METHODOLOGY

Let $F$ be the family of $c$-Lipschitz continuous functions defined on a compact domain $X$ with a compact range $Y$. In other words, $\forall f \in F$, $f : X \to Y$ and $d_Y\big(f(x_1), f(x_2)\big) \leq c \cdot d_X\big(x_1, x_2\big)$,

where $(X, d_X)$ and $(Y, d_Y)$ are metric spaces, and $c$ is the Lipschitz constant. Our goal is to find a representation algorithm that can encode a function $f \in F$ into a vector representation $\mathbf{F} \in \mathbb{C}^N$. To construct it, we will feed samples of the function mapping $\left\{ \left( x_i, f(x_i) \right) \right\}$ to the representation algorithm, which will generate the vector representation based on these samples.

We require the function representation to satisfy the following: (1) Sample distribution invariance: the function representation is "not affected" by the distribution from which the samples are collected. (2) Sample size invariance: the function representation is "not affected" by the number of samples. (3) Fixed-length representation: all functions are represented by fixed-length vectors. (4) Decodability: as new inputs query the function representation, it can reconstruct the function values.

To better formalize the heuristic expression of "not affected" in Properties 1 and 2, we introduce the definition of asymptotic sample invariance to formulate an exact mathematical expression:

**Definition 1** (Asymptotic Sample Invariance). *Let $f : X \to Y$ be the function to be encoded, $p : X \to (0, 1)$ be a probability density function (pdf) on $X$, $\{x_i\}_{i=1}^n \sim p(X)$ be $n$ independent samples of $X$. Let $\mathbf{F}_n$ be the representation computed from the samples $\{x_i, f(x_i)\}_{i=1}^n$, asymptotic sample invariance implies $\mathbf{F}_n$ converges to a limit $\mathbf{F}_\infty$ independent of the pdf $p$.*

In this definition, sample size invariance is reflected because the distance between $\mathbf{F}_m$ and $\mathbf{F}_n$ can be arbitrarily small as $m, n$ become large. Sample distribution invariance is reflected because the limit $\mathbf{F}_\infty$ does not depend on the pdf $p$, as long as $p$ is supported on the whole input space $X$.

With the problem definition above, we present our hyper-dimensional function encoding (HDFE) approach. Sec. 2.1 introduces how HDFE encodes explicit functions. Sec. 2.2 generalizes HDFE to implicit function encoding. Sec. 2.3 realizes HDFE for vector-valued function encoding. Finally, Sec. 2.4 establishes the theorems that HDFE is asymptotic sample invariant and distance-preserving. Throughout the section, we assume the functions are $c$-Lipschitz continuous. The assumption will also be explained in Section 2.4. Kindly refer to Appendix A for the table of notations.

## 2.1 Explicit Function Encoding

**Encoding**  HDFE is inspired by the methodology of hyper-dimensional computing (HDC) (Kleyko et al., 2023), where one encodes an indefinite number of data points into a fixed-length vector. The common practice is to first map the data points to a high-dimensional space and then average the data point representations in that space. The resulting superposed vector can represent the distribution of the data. Following the idea, we represent an explicit function as the superposition of its samples:

$$\mathbf{F} = \sum_i w_i \cdot E(x_i, y_i) \tag{1}$$

where $E$ maps function samples to a high-dimensional space $\mathbb{C}^N$. The question remains (a) how to design the mapping to make the vector decodable; (b) how to determine the weight of each sample $w_i$ so that the representation is sample invariant. We will answer question (a) first and leave question (b) to the iterative refinement section.

Regarding the selection of $E(x, y)$, a counter-example is a linear mapping, where the average of the function samples in the high-dimensional space will degenerate to the average of the function samples, which does not represent the function. To avoid degeneration, the encodings of the samples should not interfere with each other if they are far from each other. Specifically, if $d_X(x_1, x_2)$ is larger than a threshold $\epsilon_0$, their function values $f(x_1), f(x_2)$ may be significantly different. In this case, we want $E(x_1, y_1)$ to be orthogonal to $E(x_2, y_2)$ to avoid interference. On the other hand, if $d_X(x_1, x_2)$ is smaller than the threshold $\epsilon_0$, by the Lipschitz continuity, the distance between their function values $d_Y(f(x_1), f(x_2))$ is bounded by $c\epsilon_0$. In this case, we want $E(x_1, y_1)$ to be similar to $E(x_2, y_2)$. We call the tunable threshold $\epsilon_0$ the *receptive field* of HDFE, which will be discussed in Sec. 2.4. Denoting the similarity between vectors as $\langle \cdot, \cdot \rangle$, the requirement can be formulated as:

$$\langle E(x, y), E(x', y') \rangle \begin{cases} \approx 1 & d_X(x, x') < \epsilon_0 \\ \text{decays to 0 quickly} & d_X(x, x') > \epsilon_0 \end{cases} \tag{2}$$

In addition to avoiding degeneration, we also require the encoding to be decodable. This can be achieved by factorizing $E(x, y)$ into two components: We first map $x_i$ and $y_i$ to the high-dimensional space $\mathbb{C}^N$ through two different mappings $E_X$ and $E_Y$. To ensure equation 2 is satisfied, we require $\langle E_X(x), E_X(x') \rangle \approx 1$ when $d_X(x, x') < \epsilon_0$ and that it decays to 0 otherwise. The

property of $E_Y$ will be mentioned later in the discussion of decoding. Finally, we compute the joint embedding of $x_i$ and $y_i$ through a *binding* operation $\otimes$: $E(x_i, y_i) = E_X(x_i) \otimes E_Y(y_i)$.

We will show that the representation is decodable if the binding operation satisfies these properties:

1. commutative: $x \otimes y = y \otimes x$
2. distributive: $x \otimes (y + z) = x \otimes y + x \otimes z$
3. similarity preserving: $\langle x \otimes y, x \otimes z \rangle = \langle y, z \rangle$.
4. invertible: there exists an associative, distributive, similarity preserving operator that undoes the binding, called *unbinding* $\oslash$, satisfying $(x \otimes y) \oslash z = (x \oslash z) \otimes y$ and $(x \otimes y) \oslash x = y$.

The binding and unbinding operations can be analogous to multiplication and division, where the difference is that binding and unbinding operate on vectors and are similarity preserving.

**Decoding** With the properties of the two operations, the decoding of the function representation can be performed by a similarity search. Given the function representation $\mathbf{F} \in \mathbb{C}^N$, and a query input $x_0 \in X$, the estimated function value $\hat{y}_0$ is computed by:

$$\hat{y}_0 = \mathrm{argmax}_{y \in Y} \langle \mathbf{F} \oslash E_X(x_0), E_Y(y) \rangle \tag{3}$$

The distributive property allows the unbinding operation to be performed sample-by-sample. The invertible property allows the unbinding operation to recover the encoding of the function values: $E_X(x_i) \otimes E_Y\big(f(x_i)\big) \oslash E_X(x_0) \approx E_Y\big(f(x_i)\big) \approx E_Y\big(f(x_0)\big)$ when $d_X(x_0, x_i)$ is small. The similarity preserving property ensures that $[E_X(x_i) \oslash E_X(x_0)] \otimes E_Y(f(x_i))$ produces a vector orthogonal to $E_Y(f(x_0))$ when the distance between two samples is large, resulting in a summation of noise. The following formula illustrates the idea and Appendix E details the derivation.

$$\mathbf{F} \oslash E_X(x_0) = \sum_i w_i \cdot \big[ E_X(x_i) \otimes E_Y\big(f(x_i)\big) \oslash E_X(x_0) \big]$$

$$= \underbrace{\sum_{d(x_0, x_i) < \epsilon_0} w_i \cdot E_Y\big(f(x_i)\big)}_{\approx E_Y(f(x_0))} \quad + \quad \underbrace{\sum_{d(x_0, x_i) > \epsilon_0} w_i \cdot [E_X(x_i) \oslash E_X(x_0)] \otimes E_Y(f(x_i))}_{\text{noise, since orthogonal to } E_Y(f(x_0))}$$

After computing $\mathbf{F} \oslash E_X(x_0)$, we search for $y \in Y$ such that the cosine similarity between $E_Y(y)$ and $\mathbf{F} \oslash E_X(x_0)$ is maximized. We desire that $\frac{\partial}{\partial y}\langle E_Y(y), E_Y(y') \rangle > 0$ for all $y$ and $y'$ so that the optimization can be solved by gradient descent. See Appendix F for detailed formulation.

Since the decoding only involves measuring cosine similarity, in the last step, we normalize the function representation to achieve sample size invariance without inducing any loss:

$$\mathbf{F} = normalize\Big( \sum_i w_i \cdot \big[ E_X\big(x_i\big) \otimes E_Y\big(f(x_i)\big) \big] \Big) \tag{4}$$

**Iterative refinement for sample distribution invariance** In equation 4, we are left to determine the weight of each sample so that the representation is sample invariant. To address this, we propose an iterative refinement process to make the encoding invariant to the sample distribution. We initialize $w_i = 1$ and compute the initial function vector. Then we compute the similarity between the function vector and the encoding of each sample. We then add the sample encoding with the lowest similarity

---

**Algorithm 1** Iterative Refinement

$z_i \leftarrow E_X(x_i) \otimes E_Y(f(x_i))$ for all $i$.
$\mathbf{F} = \sum_i z_i$
**while** $\min_i \langle \mathbf{F}, z_i \rangle$ still increases **do**
    $j = \mathrm{argmin}_i \langle \mathbf{F}, z_i \rangle$
    $\mathbf{F} \leftarrow \mathbf{F} + z_j$
**end while**

---

to the function vector and repeat this process until the lowest similarity no longer increases. By doing so, the output will be *the center of the smallest ball containing all the sample encodings*. Such output is asymptotic sample invariant because the ball converges to the smallest ball containing $\cup_{x \in X}[E_X(x) \otimes E_Y(f(x))]$ as the sample size goes large, where the limit ball only depends on the function. We left the formal proof to the Appendix H.1. In Appendix I.3, we introduce a practical implementation of the iterative refinement for saving computational cost.

## 2.2 IMPLICIT FUNCTION ENCODING

Generalizing HDFE to implicit functions is fairly straightforward. Without loss of generality, we assume an implicit function is represented as $f(x) = 0$. Then it can be encoded using equation 5,

where the weights $w_x$ are determined by the iterative refinement:

$$\mathbf{F}_{f=0} = normalize\Big( \sum_{x:f(x)=0} w_x \cdot E_X(x) \Big) \tag{5}$$

The formula can be understood as encoding an explicit function $g$, where $g(x) = 1$ if $f(x) = 0$ and $g(x) = 0$ if $f(x) \neq 0$. Then by choosing $E_Y(1) = 1$ and $E_Y(0) = 0$ in equation 4, we can obtain equation 5. The formula can be interpreted in a simple way: a continuous object can be represented as the summation of its samples in a high-dimensional space.

## 2.3 VECTOR-VALUED FUNCTION ENCODING

In the previous sections, we established a theoretical framework for encoding $c$-Lipschitz continuous functions. In this section, we put this framework into practice by carefully choosing appropriate input and output mappings $E_X$, $E_Y$, the binding operator $\otimes$, and the unbinding operator $\oslash$ in equation 4. We will first state our choice and then explain the motivation behind it.

**Formulation** Let $(\mathbf{x}, y)$ be one of the function samples, where $\mathbf{x} \in \mathbb{R}^m$ and $y \in \mathbb{R}$, the mapping $E_X : X \to \mathbb{C}^N$, $E_Y : \mathbb{R} \to \mathbb{C}^N$ and the operations $\otimes$ and $\oslash$ are chosen as:

$$E_X(\mathbf{x}) := \exp\big(i \cdot \alpha \frac{\Phi\mathbf{x}}{m}\big) \qquad E_Y(y) := \exp\big(i\beta\Psi y\big) \tag{6}$$

$$E_X(\mathbf{x}) \otimes E_Y(y) := \exp\big(i \cdot \alpha \frac{\Phi\mathbf{x}}{m} + i\beta\Psi y\big)$$

$$E_X(\mathbf{x}) \oslash E_Y(y) := \exp\big(i \cdot \alpha \frac{\Phi\mathbf{x}}{m} - i\beta\Psi y\big)$$

where $i$ is the imaginary unit, $\Phi \in \mathbb{R}^{N \times m}$ and $\Psi \in \mathbb{R}^N$ are random fixed matrices where all elements are drawn from the standard normal distribution. $\alpha$ and $\beta$ are hyper-parameters controlling the properties of the mappings.

**Motivation** The above way of mapping real vectors to high-dimensional spaces is modified from Komer & Eliasmith (2020), known as fractional power encoding (FPE). We introduce the motivation for adopting this technique heuristically. In Appendix G, we elaborate on the relation between FPE and radial basis function (RBF) kernels, which gives a rigorous reason for adopting this technique.

First, the mappings are continuous, which can avoid losses when mapping samples to the embedding space. Second, the receptive field of the input mapping $E_X$ (the $\epsilon_0$ in equation 2) can be adjusted easily through manipulating $\alpha$. Fig. 2a demonstrates how manipulating $\alpha$ can alter the behavior of $E_X$. Typically, $\alpha$ has a magnitude of 10 for capturing the high-frequency component of the function. Thirdly, the decodability of the output mapping $E_Y$ can easily be achieved by selecting appropriate $\beta$ values. We select $\beta$ such that $\langle E_Y(0), E_Y(1) \rangle$ is equal to 0 to utilize the space $\mathbb{C}^N$ maximally while keeping the gradient of $\langle E_Y(y_1), E_Y(y_2) \rangle$ non-zero for all $y_1$ and $y_2$. Per the illustration in Fig. 2a, the optimal choice for $\beta$ is 2.5. Finally, the binding and unbinding operators are defined as the element-wise multiplication and division of complex vectors, which satisfy the required properties.

## 2.4 PROPERTIES OF HDFE

HDFE produces an explicit decodable representation of functions. In this section, we state a theorem on the asymptotic sample invariance, completing the claim that HDFE satisfies all four desirable properties. We study the effect of the receptive field on the behavior of HDFE. We also state that HDFE is distance-preserving and discuss the potential of scaling HDFE to high-dimensional data. We leave the proofs to Appendix H and verify the claims with empirical experiments in Appendix I. We include several empirical experiments of HDFE in Appendix I, including the cost of the iterative refinement and its practical implementation, the effectiveness of sample invariance in a synthetic regression problem, and the analysis of information loss when encoding continuous objects.

**Theorem 1** (Sample Invariance). *HDFE is asymptotic sample invariant (defined at **Definition 1**).*

HDFE being sample invariant ensures functions realized with different sampling schemes are treated invariantly. Kindly refer to Appendix H.1 and I.1 for the proof and empirical experiments.

**Theorem 2** (Distance Preserving). *Let $f, g : X \to Y$ be both c-Lipschitz continuous, then their L2-distance is preserved in the encoding. In other words, HDFE is an isometry:*

$$||f - g||_{L_2} = \int_{x \in X} |f(x) - g(x)|^2 dx \approx b - a\langle \mathbf{F}, \mathbf{G} \rangle$$

HDFE being isometric indicates that HDFE encodes functions into a organized embedding space, which can reduce the complexity of the machine learning architecture when training downstream tasks on the functions. Kindly refer to Appendix H.2 and I.2 for the proof and empirical experiment.

**Effect of receptive field**  Fig. 2 shows the reconstruction results of a 1d function $f : \mathbb{R} \to \mathbb{R}$, which demonstrates that HDFE can reconstruct the original functions given a suitable receptive field and a sufficiently large embedding space. When using a large receptive field (Fig. 2b), the high-frequency components will be missed by HDFE. When using a small receptive field (Fig. 2c), the high-frequency components can be captured, but it may cause incorrect reconstruction if the dimension of the embedding space is not large enough. Fortunately, reconstruction failures can be eliminated by increasing the dimension of the embedding space (Fig. 2d).

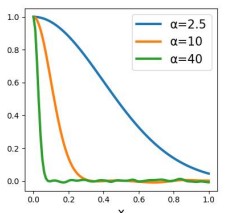 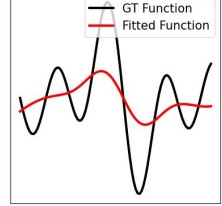 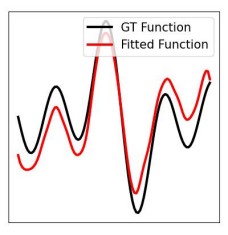 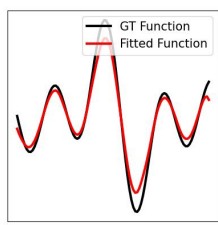

(a) Similarity between $E_X(x)$ and $E_X(0)$.

(b) Large recept. field. Dimension = 1000.

(c) Small recept. field. Dimension = 1000.

(d) Small recept. field. Dimension = 2000.

Figure 2: (a): How $\alpha$ and $\beta$ in equation 6 affects the receptive field of HDFE. (b)-(d): Functions can be reconstructed accurately given a suitable receptive field and encoding dimension. To capture the high-frequency component of the function, a small receptive field and a high dimension are required.

**Scale to high-dimensional input**  HDFE produces the function encoding based on the sparsely collected samples. Unlike mesh-grid based methods, which require a mesh-grid domain and suffer from an exponential increase in memory and computational cost as the data dimension increases, HDFE uses superposition to encode all the samples defined in the support of the function. This means the required dimensionality only depends on the size of the support, not on the data dimensionality. Even if the data dimensionality is high, HDFE can mitigate this issue as long as the data reside in a low-rank subspace. Appendix I.6 gives an empirical experiment of high-dimensional input to show the potential of HDFE to work in low-rank high-dimensional scenarios.

## 3 EXPERIMENT

In this section, we present two applications of HDFE. Sec. 3.1 showcases how HDFE can be leveraged for solving partial differential equations (PDE). This exemplifies how HDFE can enhance neural networks to receive function inputs and produce function outputs. In Sec. 3.2, we apply HDFE to predict the surface normal of point clouds. This demonstrates how HDFE can enhance neural networks to process implicit functions and extract relevant attributes.

### 3.1 PDE SOLVER

Several neural networks have been developed to solve partial differential equations (PDE), such as the Fourier neural operator (Li et al., 2020a). In this section, we compare our approach using HDFE against the current approaches and show that we achieve on-par performance. VFA does not apply to the problem since the input and output functions do not conform to the form that VFA requires.

**Architecture**  To solve PDEs using neural networks, we first encode the PDE and its solution into their vector embeddings using HDFE. Then, we train a multi-layer perceptron to map the embedding of the PDE to the embedding of its solution. The optimization target is the cosine similarity between

the predicted embedding and the true embedding. Since the embeddings are complex vectors, we adopt a Deep Complex Network (Trabelsi et al., 2017) as the architecture of the multi-layer perceptron. The details are presented in Appendix J.1. Once the model is trained, we use it to predict the embedding of the solution, which is then decoded to obtain the actual solution.

**Dataset**  We use 1d Burgers' Equation (Su & Gardner, 1969) and 2d Darcy Flow (Tek, 1957) for evaluating our method. The error is measured by the absolute distance between the predicted solution and the ground-truth solution. The benchmark (Li et al., 2020c) has been used to evaluate neural operators widely. For the 1d Burgers' Equation, it provides 2048 PDEs and their solutions, sampled at a 1d mesh grid at a resolution of 8192. For the 2d Darcy Flow, it provides 2048 PDEs and their solutions, sampled at a 2d mesh grid at a resolution of $241 \times 241$.

**Baselines**  We evaluate our HDFE against other neural network PDE-solving methods. These include: PCANN (Bhattacharya et al., 2021); MGKN: Multipole Graph Neural Operator (Li et al., 2020c); FNO: Fourier Neural Operator (Li et al., 2020a).

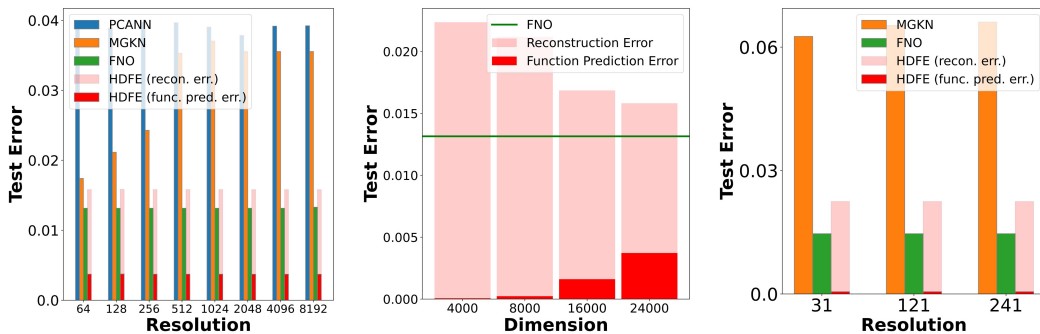

Figure 3: HDFE solves a PDE by predicting the encoding of its solution and then reconstructing at points, so the error consists of a function encoding prediction error and a reconstruction error. **Left**: Prediction error of different methods under different testing resolutions, evaluated on the 1d Burgers' equation. **Mid**: The reconstruction error (in HDFE) dominates the function encoding prediction error, while the reconstruction error can be reduced by increasing the dimensionality of the embedding. **Right**: Prediction error of different methods evaluated on 2d Darcy Flow.

*When decoding is required*, **our approach achieves $\sim 55\%$ lower prediction error than MGKN and PCANN and competitive performance to FNO**. The error of HDFE consists of two components. The first is the error arising from predicting the solution embedding, and the second is the reconstruction error arising when decoding the solution from the predicted embedding. In contrast, FNO directly predicts the solution, and hence, does not suffer from reconstruction error. If we consider both errors, HDFE achieves comparable performance to FNO. Fig. 3 shows the comparison.

On the other hand, *when decoding is not required*, **our approach achieves lower error than FNO**. Such scenarios happen frequently when we use functions only as input, for example, the local geometry prediction problem in Experiment 3.2. Despite the presence of reconstruction error, **the reconstruction error can be reduced by increasing the embedding dimension**, as shown in Figure 3 (Mid). Increasing the embedding dimension may slightly increase the function prediction error, possibly because the network is not adequately trained due to limited training data and some overfitting. We conjecture that this prediction error can be reduced with more training data.

In addition to comparable performance, **HDFE overcomes two limitations of FNO**. First, HDFE provides an explicit function representation, resolving the restriction of FNO, which only models the mappings between functions without extracting attributes from them. Second, HDFE not only works for grid-sampled functions but also for sparsely sampled functions.

## 3.2 Unoriented Surface Normal Estimation

Next, we apply HDFE to extract attributes of functions, a setting where neither neural operators nor VFA applies, because neural operators do not consume sparse samples and VFA does not encode implicit functions. We predict the unoriented surface normal from 3d point cloud input.

**Baselines** We compare our HDFE with two baselines. In the first baseline, we compare the vanilla HDFE with the PCPNet (Guerrero et al., 2018), which is a vanilla PointNet (Qi et al., 2017a) architecture. We replace the PointNet with our HDFE appended with a deep complex network (Trabelsi et al., 2017). In the second baseline, we incorporate HDFE into HSurf-Net (Li et al., 2022), which is the state-of-the-art PointNet-based normal estimator. In both settings, we compare the effect of data augmentation in the HDFE module, where we add noise to the weight of each sample when generating the patch encoding by HDFE. Kindly refer to Appendix J.2 J.3 for details.

**Dataset and metrics** We use the root mean squared angle error (RMSE) as the metrics, evaluated on the PCPNet (Guerrero et al., 2018) and FamousShape (Li et al., 2023) datasets. We compare the robustness for two types of data corruption: (1) point density: sampling subsets of points with two regimes, where *gradient* simulates the effects of distance from the sensor, and *strips* simulates local occlusions. (2) point perturbations: adding Gaussian noise to the point coordinates. Table 1 reports normal angle RMSE comparison with the baselines on PCPNet and FamousShape. Appendix K reports the ablation studies examining the effect of the receptive field size and the dimensionality.

Table 1: Unoriented normal RMSE results on datasets PCPNet and FamousShape. Replacing from PointNet to HDFE improves performance. Integrating HDFE with the SOTA estimator improves its performance. Applying data augmentation to HDFE improves its performance.

| | PCPNet Dataset (↓ means improvement) | | | | | | | FamousShape Dataset (↓ means improvement) | | | | | | |
| | Noise | | | | Density | | Average | Noise | | | | Density | | Average |
| | None | Low | Med | High | Stripe | Gradient | | None | Low | Med | High | Stripe | Gradient | |
| PCPNet Guerrero et al. (2018) | 9.64 | 11.51 | 18.27 | 22.84 | 11.73 | 13.46 | 14.58 | 18.47 | 21.07 | 32.60 | 39.93 | 18.14 | 19.50 | 24.95 |
| PCPNet - PointNet + HDFE | 9.48 | 11.05 | **17.16** | **22.53** | 11.61 | 10.19 | 13.67 | 15.66 | **17.92** | 31.24 | 38.89 | 17.26 | 14.55 | 22.59 |
| Difference | 0.16 ↓ | 0.46 ↓ | 1.11 ↓ | 0.31 ↓ | 0.12 ↓ | 3.27 ↓ | 0.91 ↓ | 2.81 ↓ | 3.15 ↓ | 1.36 ↓ | 1.04 ↓ | 0.88 ↓ | 4.95 ↓ | 2.36 ↓ |
| PCPNet - PointNet + HDFE + Aug. | **7.97** | **10.72** | 17.69 | 22.76 | **9.47** | **8.67** | **12.88** | **13.04** | 17.99 | **31.23** | **38.57** | **14.01** | **12.13** | **21.16** |
| Difference | 1.67 ↓ | 0.79 ↓ | 0.58 ↓ | 0.08 ↓ | 2.26 ↓ | 4.79 ↓ | 1.70 ↓ | 5.43 ↓ | 3.08 ↓ | 1.37 ↓ | 1.36 ↓ | 4.13 ↓ | 7.37 ↓ | 3.79 ↓ |
| HSurf-Net Li et al. (2022) | 4.30 | 8.78 | 16.15 | **21.64** | 5.18 | 5.03 | 10.18 | 7.54 | 15.56 | 29.47 | 38.61 | 7.82 | 7.44 | 17.74 |
| HSurf-Net + HDFE | 4.13 | **8.64** | **16.14** | **21.64** | 5.02 | 4.87 | 10.07 | 7.46 | **15.50** | 29.42 | 38.56 | 7.77 | 7.35 | 17.68 |
| Difference | 0.17 ↓ | 0.14 ↓ | 0.01 ↓ | 0.00 | 0.16 ↓ | 0.16 ↓ | 0.11 ↓ | 0.08 ↓ | 0.06 ↓ | 0.04 ↓ | 0.05 ↓ | 0.05 ↓ | 0.09 ↓ | 0.06 ↓ |
| HSurf-Net + HDFE + Aug. | **3.89** | 8.78 | **16.14** | 21.65 | **4.60** | **4.51** | **9.93** | **7.11** | 15.57 | **29.44** | **38.57** | **6.97** | **6.98** | **17.44** |
| Difference | 0.41 ↓ | 0.00 | 0.01 ↓ | 0.01 ↑ | 0.58 ↓ | 0.52 ↓ | 0.25 ↓ | 0.43 ↓ | 0.01 ↑ | 0.03 ↓ | 0.04 ↓ | 0.85 ↓ | 0.46 ↓ | 0.30 ↓ |

**HDFE significantly outperforms the PointNet baseline.** When processing the local patches, we replace PointNet with HDFE followed by a neural network. This replacement leads to an average reduction in error of 1.70 and 3.79 on each dataset. This is possibly because HDFE encodes the distribution of the local patch, which is guaranteed by the decodability property of HDFE. PointNet, on the other hand, does not have such guarantee. Specifically, PointNet aggregates point cloud features through a max-pooling operation, which may omit points within the point cloud and fail to adequately capture the patch's distribution. Consequently, in tasks where modeling the point cloud distribution is crucial, such as normal estimation, PointNet exhibits higher error compared to HDFE.

**HDFE, as a plug-in module, improves the SOTA baseline significantly**. HSurf-Net (Li et al., 2022), the SOTA method in surface normal estimation, introduces many features, such as local aggregation layers, and global shift layers specifically for the task. Notably, HDFE does not compel such features. We incorporate HDFE into HSurf-Net (See Appendix J.3 for details), where it leads to average error reductions of 0.25/0.30 on each dataset. Notably, such incorporation can be performed on any PointNet-based architecture across various tasks. Incorporating HDFE to other PointNet-based architectures for performance and robustness gain can be a future research direction.

**HDFE promotes stronger robustness to point density variance.** In both comparisons and both benchmarks, HDFE exhibits stronger robustness to point density variation than its PointNet counterpart, especially in the Density-Gradient setting (error reduction of 4.79/7.37/0.52/0.46). This shows the effectiveness of the HDFE's sample invariance property and the embedding augmentation. Sample invariance ensures a stable encoding of local patches when the point density changes. The embedding augmentation is a second assurance to make the system more robust to density variation.

## 4 RELATED WORK

### 4.1 MESH-GRID-BASED FRAMEWORK

These methods operate on continuous objects (functions) defined on mesh grids with arbitrary resolution. They enjoy discretization invariance, but they do not receive sparse samples as input.

**Fourier transform** (FT) can map functions from their time domains to their frequency domains. By keeping a finite number of frequencies, FT can provide a vector representation of functions. 1D-

FT has been a standard technique for signal processing (Salih, 2012) and 2D-FT has been used for image processing (Jain, 1989). FT is also incorporated into deep learning architectures (Fan et al., 2019; Sitzmann et al., 2020). However, FT is not scalable since the $n$-dimensional Fourier transform returns an $n$-dimensional matrix, which is hard to process when $n$ gets large.

**Neural operator** is a set of toolkits to model the mapping between function spaces. The technique was pioneered in DeepONet (Lu et al., 2019) and a series of tools were developed (Li et al., 2021; 2020b;c; Guibas et al., 2021; Kovachki et al., 2021) for studying the problem. The most well-known work is the Fourier Neural Operator (FNO) (Li et al., 2020a). The approach showed promising accuracy and computational efficiency. Though proposed in 2020, FNO is still the first choice when mapping between function spaces (Wen et al., 2023; Renn et al., 2023; Gopakumar et al., 2023). Despite their success, neural operators lack explicit function representations and their application is limited to mappings between function spaces.

### 4.2 SPARSE FRAMEWORK

These methods, including HDFE, work with sparse samples from continuous objects.

**PointNet** (Qi et al., 2017a) is a neural network architecture for processing point cloud data. It uses multi-layer perceptrons to capture local features of every point and then aggregates them into a global feature vector that's invariant to the order of the input points. PointNet and its variation (Zaheer et al., 2017; Qi et al., 2017b; Joseph-Rivlin et al., 2019; Yang et al., 2019; Zhao et al., 2019; Duan et al., 2019; Yan et al., 2020) have been widely applied to sparse data processing, for example, for object classification (Yan et al., 2020; Lin et al., 2019), semantic segmentation (Ma et al., 2020; Li et al., 2019), and object detection (Qi et al., 2020; Yang et al., 2020) with point cloud input. However, PointNet does not produce a decodable representation. Specifically, after encoding a point cloud with PointNet, it is difficult to decide whether a point is drawn from the point cloud distribution. Besides, PointNet is also sensitive to perturbations in the input point cloud.

**Kernel methods** (Hofmann et al., 2008) are a type of machine learning algorithm that transforms data into a higher-dimensional feature space via a kernel function, such as the radial basis function (RBF) (Cortes & Vapnik, 1995), which can capture nonlinear relationships. Though kernel methods can predict function values at any query input (i.e. decodable) and the prediction is invariant to the size and distribution of the training data, kernel methods do not produce an explicit representation of functions, so they are only used for fitting functions but not processing or predicting functions.

**Vector function architecture** VFA (Frady et al., 2021) encodes a function of the form $f(x) = \sum_k \alpha_k \cdot K(x, x_k)$ into a vector, where $K : X \times X \to \mathbb{R}$ is a kernel defined on the input space. VFA and HDFE share a similar high-level idea. They both map the samples to high-dimensional space and compute the weighted average in that space. However VFA determines the weight by relying on the assumption of the functional form. HDFE, on the other hand, uses iterative refinement to solve the weights. The iterative refinement coupled with the binding operation relaxes the assumption required by VFA and enables encoding functions across a larger class of inputs. In addition, VFA is limited to empirical experiments such as non-linear regression and density estimation, without practical applications. In comparison, HDFE is demonstrated to be applicable to real-world problems.

## 5 CONCLUSION

We introduced Hyper-Dimensional Function Encoding (HDFE), which constructs vector representations for continuous objects. The representation, without any training, is sample invariant, decodabile, and isometric. These properties position HDFE as an interface for the processing of continuous objects by neural networks. Our study demonstrates that the HDFE-based architecture attains significantly reduced errors compared to PointNet-based counterparts, especially in the presence of density perturbations. This reveals that HDFE presents a promising complement to PointNet and its variations for processing point cloud data. Adapting HDFE (e.g. imposing rotational invariance to HDFE) to tasks like point cloud classification and segmentation offers promising avenues for exploration. Still, HDFE does possess limitations in encoding capacity. For functions defined over large domains or highly non-linear functions, HDFE can experience underfitting. The exploration of techniques to enhance HDFE's capacity remains promising research. Regardless, HDFE already shows strong applicability in low-dimensional (1D, 2D, 3D) inputs.

## 6 ACKNOWLEDGEMENT

The support of NSF under awards OISE 2020624 and BCS 2318255, and ARL under the Army Cooperative Agreement W911NF2120076 is greatly acknowledged. The in-depth discussions with Denis Kleyko, Paxon Frady, Bruno Olshausen, Christopher Kymn, Friedrich Sommer, Pentti Kanerva significantly improved the manuscript and we are grateful.

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

# APPENDIX

## A   NOTATION TABLE

Table 2: Table of Notation

<hr>

### Notations of Continuous Functions

| | | |
|---:|:---:|:---|
| $F$ | $\triangleq$ | family of $c$-Lipschitz continuous functions |
| $c$ | $\triangleq$ | Lipschitz constant of the continuous function |
| $X$ | $\triangleq$ | input space, a bounded domain of the function |
| $Y$ | $\triangleq$ | output space, a bounded range of the function |
| $d_X$ | $\triangleq$ | distance metrics in the input space $X$ |
| $d_Y$ | $\triangleq$ | distance metrics in the output space $Y$ |
| $\{(x_i, f(x_i)\}$ | $\triangleq$ | collection of function samples. |
| $m$ | $\triangleq$ | dimensionality of the input space $X$. |

### Notations of HDFE

| | | |
|---:|:---:|:---|
| $\epsilon_0$ | $\triangleq$ | receptive field of HDFE. |
| $\mathbf{F}$ | $\triangleq$ | $N$-dimensional complex vector. Encoding of the explicit function $f$. |
| $\mathbf{F}_{f=0}$ | $\triangleq$ | $N$-dimensional complex vector. Encoding of the implicit function $f(x) = 0$. |
| $N$ | $\triangleq$ | dimensionality of the function encoding. |
| $E_X, E_Y$ | $\triangleq$ | the input and output mapping from $X, Y$ to the encoding space $\mathbb{C}^N$. |
| $\otimes$ | $\triangleq$ | binding operation. |
| $\oslash$ | $\triangleq$ | unbinding operation. |
| $\langle \cdot, \cdot \rangle$ | $\triangleq$ | cosine similarity between two complex vectors. |

<hr>

## B   POINTNET AS FUNCTION ENCODER

In this section, we present empirical evidence highlighting PointNet's limitations in generating decodable function encodings. Our designed pipeline, following the popular style of training reconstruction networks, demonstrates PointNet's inability to produce even reasonable reconstructions. Consequently, we infer that without substantial modifications, PointNet is unlikely to acquire the necessary capability for effective function encoding.

Specifically, we generate random functions by

$$f(x) = \frac{1}{2} + \frac{1}{8} \sum_{k=1}^{4} a_k \sin(2\pi k x)$$

where $a_k \sim Uniform(0, 1)$ are the parameters controlling the generation and $f(x) \in (0, 1)$. We randomly sample $\{x_i, y_i\}_{i=1}^{5000}$ where $y_i = f(x_i) + \epsilon_i$, $x_i \sim Uniform(0, 1)$ and $\epsilon_i$ is white noise with variance 1e-4. Then we stack $x_i$ and $y_i$ into a $(5000, 2)$ matrix and feed it to a standard PointNet to generate a 1024-dimensional vector. After encoding the function samples, we introduce a decoder that retrieves the function values from the encoding when receiving function inputs. We optimize the PointNet encoder and the decoder by minimizing the reconstruction loss. The procedure can be summarized as:

$$\mathbf{F} = \texttt{PointNet}\big(\{(x_i, y_i)\}_{i=1}^{n}\big), \quad \hat{y}_i = \texttt{Decoder}(\mathbf{F}, x_i), \quad \mathcal{L} = \sum_{i=1}^{n} |y_i - \hat{y}_i|^2$$

The decoder is designed as followed: $x_i$ is lifted to a 1024-dimensional vector $\tilde{\mathbf{x}}_i$ through sequential layers of `Linear(1,64)`, `Linear(64,128)`, `Linear(128,1024)` with `BatchNorm` and

`ReLU` inserted between the linear layers. The retrieved function value $\hat{y}_i$ is computed by the dot product between $\mathbf{F}$ and $\tilde{\mathbf{x}}_i$. Figure 4 (**Left**) plots the training curve and Figure 4 (**Right**) visualizes the reconstruction, which shows PointNet fails to learn a decodable representation.

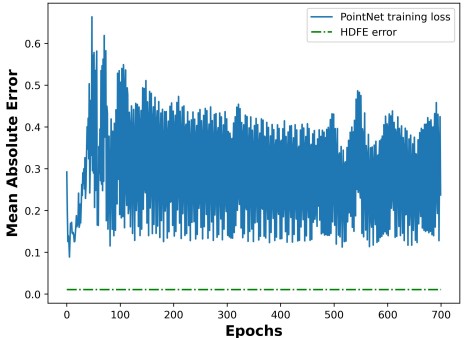 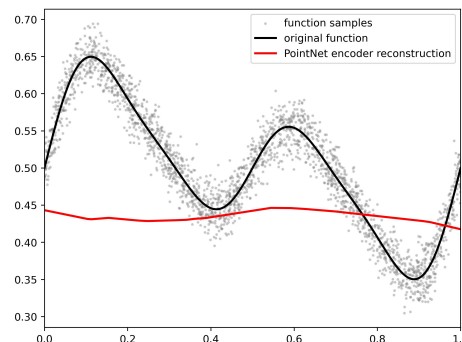

Figure 4: PointNet fails to learn an effective function encoder. **Left**: Training curve of the PointNet function encoder. The error is still significantly higher than HDFE though training for 700 epochs. **Right**: The PointNet function encoder does not produce a reasonable reconstruction.

## C VECTOR FUNCTION ARCHITECTURE

In this section, we will give a brief introduction of VFA and its relation and comparison with HDFE.

**Vector Function Architecture (VFA)** A function of the form

$$f(x) = \sum_k \alpha_k \cdot K(x, x_k) \tag{7}$$

is represented by $\mathbf{F} = \sum_k \alpha_k \cdot E(x_k)$, where the kernel $K(\cdot, \cdot)$ and the encoder $E(\cdot)$ satisfy $K(x, y) = \langle E(x), E(y) \rangle$. Given the function encoding $\mathbf{F}$, the function value at a query point $x_0$ can be retrieved by $\hat{f}(x_0) = \langle \mathbf{F}, E(x_0) \rangle$.

VFA satisfies the four desired properties: VFA can encode an explicit function into a fixed-length vector, where the encoding is sample invariant and decodable. Besides, the function encoding will preserve the overall similarity of the functions: $\langle \mathbf{F}, \mathbf{G} \rangle = \int_x f(x)g(x)dx$. Note that when encoding multiple functions, the kernel $K(\cdot, \cdot)$ and the encoder $E(\cdot)$ must be the same. Otherwise, the function encoding generated by VFA will be meaningless.

However, it is a strong assumption that a function can be approximated by equation 7. In other words, under a fixed selection of $K(\cdot, \cdot)$ and $E(\cdot)$, equation 7 can only approximate a small set of functions. There are a large set of functions that cannot be approximated by equation 7. In Fig. 5, we show a failure case of VFA. We generate a function $f(x) = 0.1 + \sum_{k=1}^{10} \alpha_k \cdot K(x, x_k)$, where $x_k \sim N(0, 1)$, $\alpha_k \sim Uniform(0, 1)$. $K(x, y) = \exp[-20(x - y)^2]$ is an RBF kernel. We generate 1000 function samples and add white noise to the output values to form the training set. The experiment shows that VFA fails to reconstruct the function but HDFE succeeds.

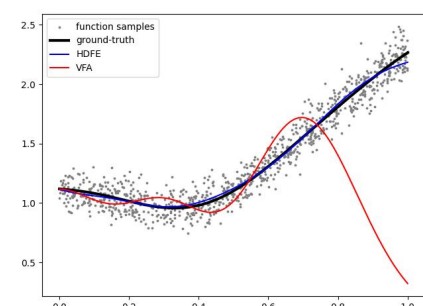

Figure 5: VFA fails to reconstruct the function but HDFE succeeds.

In this paper, HDFE is applicable for all $c$-Lipchitz function, without compromising the four desired properties. Thanks to the improvement, we can apply function encoding to real-world applications. The first application, PDE solving, cannot be solved by VFA because the input and output functions cannot be approximated by equation 7. The second application, local geometry prediction, has to deal with implicit function encoding, which cannot be solved by VFA either because VFA only encodes explicit functions.

## D  SUITABLE INPUT TYPES FOR HDFE

We mentioned HDFE can encode Lipschitz functions. In this section, we will discuss several data types that exhibits Lipschitz continuity, making them well-suited as inputs for HDFE. This enumeration is not exhaustive of all potential suitable inputs for HDFE. Our objective is to furnish a conceptual understanding of the characteristics that define a suitable input, thereby guiding future considerations in this domain.

**Point Cloud Local Geometry** In many 3D vision problems, point cloud local features are important, such as point cloud registration (Huang et al., 2021), scene matching (Han et al., 2023). The local geometry around a point in a point cloud is usually a continuous surface, which inherently exhibits Lipschitz continuity. This characteristic makes HDFE suitable for encoding this type of data.

**Meteorological Measurements** In the field of meteorology, machine learning plays a pivotal role in various applications, including pollutant estimation (Lu et al., 2020) and weather forecasting (Chen et al., 2022). A standard practice in these applications involves inputting meteorological measurements from neighboring areas (for instance, windows of 100 km$^2$) into machine learning models. HDFE is a suitable tool for encoding these neighboring meteorological measurements, primarily due to the inherent physical constraints that ensure these measurements do not rapidly change over time and space.

**Time Series Data**: Some time series data, especially in fields like event logging (Chen et al., 2021) or network traffic (Abbasi et al., 2021), are sparse in nature. Such sparse events can be treated as implicit functions and therefore HDFE is well-suited for it.

## E  DERIVATION OF DECODING

In this section, we complete the details why $\mathbf{F} \oslash E_X(x_0)$ can be decomposed into the summation of $E_Y(f(x_0))$ and noises. This is an immediate outcome from the similarity preserving property. When $d_X(x_0, x_i)$ is large, $\langle E_X(x_0) \otimes E_Y(f(x_0)), E_X(x_i) \otimes E_Y(f(x_i)) \rangle \approx 0$. Since the unbinding operation is similarity preserving, we have $\langle E_Y(f(x_0)), E_X(x_i) \otimes E_Y(f(x_i)) \oslash E_X(x_0) \rangle \approx 0$. Therefore, $\mathbf{F} \oslash E_X(x_0)$ can be decomposed into $E_Y(f(x_0))$ and the sum of vectors that are orthogonal to it. Consequently, when we search for the $y$ to maximize the $\langle \mathbf{F} \oslash E_X(x_0), E_Y(y) \rangle$, those orthogonal vectors will not bias the optimization.

## F  GRADIENT DESCENT FOR DECODING FUNCTION ENCODING

In equation 3, HDFE decodes the function encoding by a similarity maximization. Given the function encoding $\mathbf{F} \in \mathbb{C}^N$, and a query point $x_0 \in X$, the function value $\hat{y}_0$ is reconstructed by:

$$\hat{y}_0 = \operatorname{argmax}_{y \in Y} \langle \mathbf{F} \oslash E_X(x_0), E_Y(y) \rangle$$

When $E_X$ and $E_Y$ are chosen as the fractional power encoding (equation 6), the optimization can be solved by gradient descent. In this section, we detail the gradient descent formulation. We assume the output space $Y = \mathbb{R}$.

By setting $E_Y$ as the fractional power encoding, the optimization can be rewritten as:

$$\hat{y}_0 = \operatorname{argmax}_{y \in (0,1)} \langle \mathbf{F} \oslash E_X(x_0), \exp(i\Psi y) \rangle^1$$

where $\Psi \in \mathbb{R}^N$ is a random fixed vector where all elements are drawn from the normal distribution. Since $\mathbf{F} \oslash E_X(x_0)$ is a constant vector, the optimization can be further simplified as:

$$\hat{y}_0 = \operatorname{argmax}_{y \in (0,1)} \langle \mathbf{z}, \exp(i\Psi y) \rangle \tag{8}$$

where $\mathbf{z} = \mathbf{F} \oslash E_X(x_0) \in \mathbb{C}^N$. We write $\mathbf{z}$ into its polar form:

$$\mathbf{z} = \left[ a_1 e^{i\theta_1}, a_2 e^{i\theta_2}, \cdots, a_N e^{i\theta_N} \right]$$

---

[1] In equation 6, $E_Y(y) = \exp(i\beta\Psi y)$. Since $\beta$ and $n$ are two constant numbers, we rewrite $\beta\Psi$ as $\Psi$ for notation purpose.

where $a_k \in [0, +\infty)$ and $\theta_k \in [0, 2\pi)$. We first simplify equation 8 and then compute its gradient with respect to $y$.

$$\langle \mathbf{z}, \exp(i\Psi y) \rangle = \frac{1}{N^2} ||\bar{\mathbf{z}} \cdot \exp(i\Psi y)||^2$$

$$= \frac{1}{N^2} \Big|\Big| \sum_{k=1}^{N} a_k e^{i\theta_k} e^{-i\Psi_k y} \Big|\Big|^2$$

$$= \frac{1}{N^2} \sum_{p=1}^{N} \sum_{q=1}^{N} a_p a_q e^{i\left[(\theta_p - \theta_q) - (\Psi_p - \Psi_q)y\right]}$$

$$= \frac{1}{N^2} \sum_{p=1}^{N} \sum_{q=1}^{N} a_p a_q \cos\left[(\theta_p - \theta_q) - (\Psi_p - \Psi_q)y\right]$$

The gradient can be computed easily by taking the derivative:

$$\frac{d}{dy}\langle \mathbf{z}, \exp(i\Psi y) \rangle = \frac{1}{N^2} \sum_{p=1}^{N} \sum_{q=1}^{N} a_p a_q (\Psi_p - \Psi_q) \sin\left[(\theta_p - \theta_q) - (\Psi_p - \Psi_q)y\right] \qquad (9)$$

In practice, $N$ can be a large number like 8000. Computing the gradient by equation 9 can be expensive. Fortunately, since $\Psi$ is a random fixed vector, where all elements are independent of each other, we can unbiasedly estimate the gradient by sampling a small number of entries in the vector. The decoding can be summarized by the pseudo-code below.

---
**Algorithm 2** Gradient Descent for Decoding Function Encoding
---

Input: $\mathbf{F}$, $x_0$, $E_X$, $\Psi$       $\triangleright$ $\mathbf{F}$ is the function encoding. $x_0$ is the query point.
      $\triangleright$ $E_X$ is the input mapping. $\Psi$ is the parameter in $E_Y$.
$\mathbf{z} = \mathbf{F} \oslash E_X(x_0)$
$\mathbf{z} = \left[a_1 e^{i\theta_1}, a_2 e^{i\theta_2}, \cdots, a_N e^{i\theta_N}\right]$       $\triangleright$ Convert $\mathbf{z}$ into its polar form.
**while** $\langle \mathbf{z}, \exp(i\Psi y) \rangle$ still increases **do**
     Randomly select a subset of $[0, N) \cap \mathbb{Z}$. Denote as $S$.       $\triangleright$ We choose $|S| = 500$.
     $g = |S|^{-2} \cdot \sum_{p,q \in S} a_p a_q (\Psi_p - \Psi_q) \sin\left[(\theta_p - \theta_q) - (\Psi_p - \Psi_q)y\right]$
     $y \leftarrow y - \alpha \cdot g$       $\triangleright$ $\alpha$ is the learning rate.
**end while**

---

## G    RELATION BETWEEN FPE AND RBF KERNEL

In the main paper, we mention that we adopt fractional power encoding (FPE) as the mapping for the input and output space. In this section, we explain the rational behind our choice by revealing its close relationship between the radial basis function (RBF) kernel. For explanation purpose, we discuss the single-variable case. It is straight-forward to generalize to multi-variable scenarios.

RBF kernel is a similarity measure defined as $K(x, y) = \exp\left(-\gamma(x - y)^2\right)$. The feature space of the kernel has an infinite number of dimensions:

$$\exp\left(-\gamma(x - y)^2\right) = \sum_{k=0}^{\infty} \frac{(-1)^k}{(2k)!} \gamma^{2k} (x - y)^{2k} = \langle \phi(x), \phi(y) \rangle$$

where $\phi(x) = e^{-\gamma x^2} \left[1, \sqrt{\frac{\gamma}{1!}}x, \sqrt{\frac{\gamma^2}{2!}}x^2, \sqrt{\frac{\gamma^3}{3!}}x^3, \cdots\right]$.

We will then show a theorem that tells that FPE maps real numbers into finite complex vectors, where the similarity between the vectors can approximate a heavy-tailed RBF kernel. However, the feature space of the heavy-tailed RBF kernel still has an infinite number of dimensions and therefore inherits all the blessings of the RBF kernel. Notably, FPE provides a finite encoding, while approximating an infinite dimensional feature space.

**Theorem 3.** *Let* $x, y \in \mathbb{R}$, $E(x) = e^{i\gamma\Theta x}$, *where* $\Theta \in \mathbb{R}^N$ *and* $\forall \theta \in \Theta, \theta \sim \mathcal{N}(0, \frac{1}{2})$, *then as* $N \to \infty$,

$$\langle E(x), E(y) \rangle = \langle e^{i\gamma\Theta x}, e^{i\gamma\Theta y} \rangle \to \sum_{k=0}^{\infty} \frac{(-1)^k}{(2k)!!}\gamma^{2k}(x-y)^{2k} = \langle \phi(x), \phi(y) \rangle$$

*where* $\phi(x) = e^{-\gamma x^2}\left[ 1, \sqrt{\frac{\gamma}{1!!}}x, \sqrt{\frac{\gamma^2}{2!!}}x^2, \sqrt{\frac{\gamma^3}{3!!}}x^3, \cdots \right]$.

*Proof.*

$$\langle e^{i\gamma\Theta x}, e^{i\gamma\Theta y} \rangle = \frac{1}{N^2}|| \exp(i\gamma\Theta x) \cdot \exp(-i\gamma\Theta y)||^2$$

$$= \frac{1}{N^2}\left|\left| \sum_{k=1}^{N} e^{i\gamma\theta_k(x-y)} \right|\right|^2$$

$$= \frac{1}{N^2}\left[ \sum_{p=1}^{N} 1 + \sum_{p \neq q} e^{i\gamma(\theta_p - \theta_q)(x-y)} \right]$$

$$= \frac{1}{N} + \frac{1}{N^2}\sum_{p \neq q} \cos\left[ \gamma(\theta_p - \theta_q)(x-y) \right]$$

$$= \frac{1}{N} + \frac{1}{N^2}\sum_{p \neq q}\sum_{k=0}^{\infty}(-1)^k \frac{\left[ \gamma(\theta_p - \theta_q)(x-y) \right]^{2k}}{(2k)!}$$

$$= \frac{1}{N} + \sum_{k=0}^{\infty}(-1)^k \left[ \sum_{p \neq q} \frac{(\theta_p - \theta_q)^{2k}}{N^2} \right]\frac{\gamma^{2k}(x-y)^{2k}}{(2k)!}$$

$$\to \sum_{k=0}^{\infty}(-1)^k(2k-1)!!\frac{\gamma^{2k}(x-y)^{2k}}{(2k)!}$$

$$= \sum_{k=0}^{\infty}\frac{(-1)^k}{(2k)!!}\gamma^{2k}(x-y)^{2k}$$

Since $\theta_p, \theta_q \sim \mathcal{N}(0, \frac{1}{2})$, $\theta_p - \theta_q \sim \mathcal{N}(0, 1)$, and therefore, $\mathbb{E}[(\theta_p - \theta_q)^{2n}] = (2n-1)!!$. So $\sum_{p \neq q}\frac{(\theta_p - \theta_q)^{2n}}{N^2}$ converges to $(2n-1)!!$. $\qquad\square$

## H  PROOF OF THE HDFE'S PROPERTIES

### H.1  ASYMPTOTIC SAMPLE INVARIANCE

In the main paper, we claim that the iterative refinement (Algorithm 1) will converge to *the center of the smallest ball containing all the sample encodings* and therefore, HDFE leads to an asymptotic sample invariant representation. In this section, we detail the proof of the argument. To facilitate the understanding, Figure 6 sketches the proof from a high-level viewpoint. Recall the definition of asymptotic sample invariance (definition 1):

**Definition** (Asymptotic Sample Invariance). *Let* $f : X \to Y$ *be the function to be encoded,* $p : X \to (0, 1)$ *be a probability density function (pdf) on* $X$, $\{x_i\}_{i=1}^n \sim p(X)$ *be* $n$ *independent samples of* $X$. *Let* $\mathbf{F}_n$ *be the representation computed from the samples* $\{x_i, f(x_i)\}_{i=1}^n$, *asymptotic sample invariance implies* $\mathbf{F}_n$ *converges to a limit* $\mathbf{F}_\infty$ *independent of the pdf* $p$.

*Proof.* We begin by showing the iterative refinement converges to

$$\mathbf{F}_n = \text{argmax}_{||z||=1} \min_{i=1}^{n}\langle z, E(x_i, f(x_i)) \rangle \qquad (10)$$

where $E(x, f(x))$ is defined at equation 6 in the original paper. It maps a function sample to a high-dimensional space $\mathbb{C}^N$.

Figure 6: Proof of asymptotic sample invariance (overview). $Ball(S)$ and $Ball(S_n)$ are the smallest ball containing $S$ and $S_n$. As $n \to \infty$, the Hausdorff distance between the two balls goes to zero with probability one. From elementary geometry, $||center(Ball(S_n)) - center(Ball(S))|| \leq d_H(Ball(S_n), Ball(S))$. So the distance between the centers of the two balls goes to 0.

To show the convergence of the iterative refinement, it follows from the gradient descent: since $\nabla[-\min_i\langle z, E(x_i, f(x_i))\rangle] = -\text{argmin}_i\langle z, E(x_i, f(x_i))\rangle$, the gradient descent is formulated as $z \leftarrow z + \alpha \cdot \text{argmin}_i\langle z, E(x_i, f(x_i))\rangle$, which aligns with the iterative refinement in the paper.

Then we will prove equation 10 produces a sample invariant encoding by proving $\mathbf{F}_n$ converges to

$$\mathbf{F}_\infty = \text{argmax}_{||z||=1} \min_{x \in X}\langle z, E(x, f(x))\rangle \tag{11}$$

Throughout the proof, we use the following definitions:

$$
\begin{aligned}
S \subset \mathbb{C}^N &\triangleq \bigcup_{x \in X} E(x, f(x)) \\
S_n \subset \mathbb{C}^N &\triangleq \bigcup_{i=1}^n E(x_i, f(x_i)) \\
||\cdot|| &\triangleq \text{L2-norm of a complex vector.} \\
d_H &\triangleq \max_{q \in Q}\min_{p \in P}||p - q||. \text{ Hausdorff distance between two compact} \\
& \quad \text{sets } P \subset Q \subset \mathbb{C}^N. \\
Ball(P) &\triangleq \text{the smallest solid ball that contains the compact set } P. \\
center(Ball(\cdot)) &\triangleq \text{center of the ball.}
\end{aligned}
$$

Recall from equation 6, $E(x, y) = \mathcal{F}^{-1}(e^{i(\Phi x + \Psi y)})$, so $||E(x, y)|| = 1$ for all $x$ and $y$. Since $||z_1 - z_2||^2 = ||z_1||^2 + ||z_2||^2 - 2\langle z_1, z_2\rangle$, we have $\langle z, E(x, y)\rangle = 1 - \frac{1}{2}||z - E(x, y)||^2$ if $||z||^2 = 1$. Therefore, equation 10 is equivalent to

$$\mathbf{F}_n = \text{argmin}_{||z||=1} \max_{i=1}^n ||z - E(x_i, f(x_i))|| \tag{12}$$

Note that equation 12 implies $\mathbf{F}_n$ **is the center of the smallest ball containing** $S_n$:

$$\mathbf{F}_n = center(Ball(S_n)) \tag{13}$$

because if we were to construct balls containing $S_n$ with a center $\mathbf{F}' \neq \mathbf{F}_n$, the radius of the ball must be larger than the radius of $Ball(S_n)$.

**When $n \to \infty$, the Hausdorff distance between $Ball(S_n)$ and $Ball(S)$ goes to 0 with probability one.** First, it is easy to see that $d_H(Ball(S_n), Ball(S))$ is a decreasing sequence and is positive, so the limit exists. Assume the limit is strictly positive, then there exists a point $p' \in S$ such that $\min_{q \in S_n} ||p' - q|| > c$ for some constant $c > 0$ as $n \to \infty$. This means no sample is drawn from the ball $B_c(p')$. This is contradictory to the definition of $p : X \to (0, 1)$: $p$ is positive over the input space $X$.

Finally, we conclude by $||\mathbf{F}_n - \mathbf{F}_\infty|| \leq d_H(Ball(S_n), Ball(S))$. Since $Ball(S_n) \subset Ball(S)$, from elementary geometry, if $A \subset B$ are two balls, then $||center(B) - center(A)|| \leq radius(B) - radius(A) \leq d_H(B, A)$. Therefore, we have $||center(Ball(S_n)) - center(Ball(S))|| \leq d_H(Ball(S_n), Ball(S))$. Therefore, $||\mathbf{F}_n - \mathbf{F}_\infty|| \leq d_H(Ball(S_n), Ball(S))$, which decays to 0 as $n \to \infty$. $\qquad \square$

## H.2 ISOMETRY

In this section, we complete the proof that HDFE is an isometry.

**Theorem.** *Let $f, g : X \to Y$ be both c-Lipschitz continuous, then their L2-distance is preserved in the encoding. In other words, HDFE is an isometry:*

$$||f - g||_{L_2} = \int_{x \in X} \big| f(x) - g(x) \big|^2 dx \approx b - a\langle \mathbf{F}, \mathbf{G} \rangle$$

**Lemma 4.** $\langle x \otimes y, x \otimes z \rangle = \langle y, z \rangle.$

*Proof.* Let $x = e^{i\mathbf{x}}, y = e^{i\mathbf{y}}, z = e^{i\mathbf{z}},$

$$\begin{aligned}
\langle x \otimes y, x \otimes z \rangle &= \langle e^{i(\mathbf{x}+\mathbf{y})}, e^{i(\mathbf{x}+\mathbf{z})} \rangle \\
&= e^{i(\mathbf{x}+\mathbf{y})} \cdot e^{-i(\mathbf{x}+\mathbf{z})} \\
&= \langle e^{i\mathbf{y}}, e^{i\mathbf{z}} \rangle \\
&= \langle y, z \rangle
\end{aligned}$$

$\square$

*Proof.*

$$\begin{aligned}
\langle \mathbf{F}, \mathbf{G} \rangle &= \int_x \int_{x'} \langle E_X(x) \otimes E_Y(f(x)), E_X(x') \otimes E_Y(g(x')) \rangle dx' dx \\
&= \int_{|x-x'|<\epsilon} \langle E_X(x) \otimes E_Y(f(x)), E_X(x') \otimes E_Y(g(x')) \rangle dx' dx \\
&\quad + \int_{|x-x'|>\epsilon} \langle E_X(x) \otimes E_Y(f(x)), E_X(x') \otimes E_Y(g(x')) \rangle dx' dx \\
&= \int_x \langle E_X(x) \otimes E_Y(f(x)), E_X(x) \otimes E_Y(g(x)) \rangle dx + noise \\
&\approx \int_x \langle E_Y(f(x)), E_Y(g(x)) \rangle dx \qquad \text{by Lemma 4.} \\
&= \int_x \sum_{k=0}^{\infty} \frac{(-1)^k}{(2k)!!} \beta^{2k} (f(x) - g(x))^{2k} dx \qquad \text{by Theorem 3} \\
&\approx b - a \int_x |f(x) - g(x)|^2 dx \qquad \text{by taking the first and second order terms}
\end{aligned}$$

The second line holds because when $d_X(x, x')$ is larger than the receptive field $\epsilon_0$, $E_X(x)$ and $E_X(x')$ will be orthogonal, so the similarity between $E_X(x) \otimes E_Y(f(y))$ and $E_X(x') \otimes E_Y(g(y))$ will be close to zero and they will be summed as noise. $\square$

## I EMPIRICAL EXPERIMENT OF HDFE

In this section, we verify the properties claimed in Sec. 2.4 with empirical experiments.

### I.1 SAMPLE INVARIANCE

In Fig. 7, we demonstrate that the function encoding produced by HDFE remains invariant of both the sample distribution and sample density. Specifically, we sample function values from three distinct input space distributions, namely left-skewed, right-skewed, and uniform distribution, each with sample sizes of either 5000 or 1000. We then calculate the similarity between the function vectors generated from these six sets of function samples. Before tuning the function vectors, the representation is influenced by the sample distributions (Fig. 7 Mid). However, after the tuning process, the function vector becomes immune to the sample distribution (Fig. 7 Right).

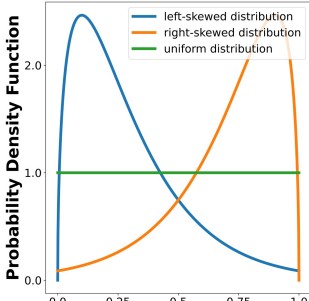 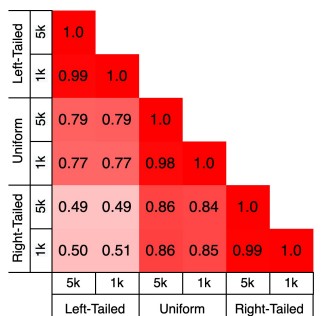 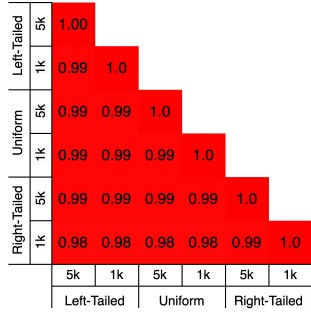

Figure 7: HDFE is invariant of sample distribution and sample size. **Left**: Three distributions where the function samples are drawn from. For each distribution, the sample size is either 5000 or 1000. **Mid, Right**: Similarity among the function vectors generated by the six sets of function samples, before and after the function vector tuning process, respectively.

## I.2 ISOMETRY

In Fig. 8, we generate pairs of random functions and compute their function encodings through HDFE. We plot the L2-distance between the functions and the similarity between their encodings. We discover a strong correlation between them. This coincides the isometric property claimed in Theorem 2.

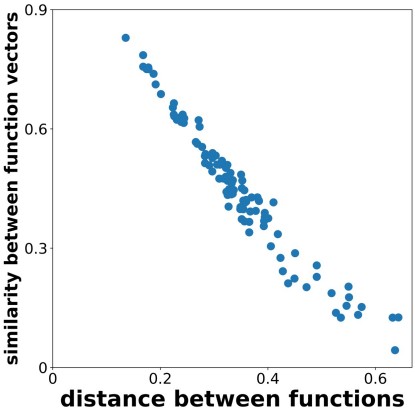

Figure 8: HDFE is a distance preserving transformation. The L2-distance between functions is proportional to the negative similarity between their encodings.

## I.3 PRACTICAL CONSIDERATION OF ITERATIVE REFINEMENT

In Algorithm 1, we propose one implementation of iterative refinement, which iteratively adds the sample encoding that has the minimum similarity with the function encoding. This implementation is a conservative implementation that guarantees asymptotic sample invariance. However, in practical applications, strict sample invariance may not be necessary. For example, achieving a similarity threshold of 0.99 when constructing with different samples might not be required. This relaxation is viable because the downstream neural network possesses an inherent ability to handle some level of inconsistency. Therefore, we consider a practical adaptation of Algorithm 1 by introducing slight modifications to accommodate these real-world considerations.

**One-Shot Refinement** In Algorithm 1, the motivation of the iterative refinement is to balance the weights between dense and sparse samples. By iterative refinement, we adjust the function encoding so that the sparse samples also contribute to the encoding. Such motivation can be achieved by another cheaper one-shot refinement. After obtaining the initial function encoding by averaging the sample encoding, we compute the similarity between this initial encoding and all the sample encodings. The similarity can serve as a rough estimation of the sample density at a particular point.

Therefore, if we were to balance the weights between dense and sparse samples, we can simply recompute the weights by the inverse estimated density. Algorithm 3 illustrates the procedure.

---

**Algorithm 3** One-Shot Refinement

$z_i \leftarrow E_X(x_i) \otimes E_Y(f(x_i))$ for all $i$.
$\mathbf{F} = \sum_i z_i$
**for** i **do**
    $w_i = \langle \mathbf{F}, z_i \rangle$                       ▷ $w_i$ is an estimation of the density at $(x_i, f(x_i))$.
    $w_i = \max(\epsilon, w_i)$                            ▷ Ensure numerical stability.
    $w_i = w_i^{-1} / \sum_{j=1}^n w_j^{-1}$                   ▷ Compute inverse density.
**end for**
$\mathbf{F} = \sum_i w_i \cdot z_i$

---

**Although one-shot refinement is not strictly sample distribution invariant, it is a quite good approximation of the sample invariant function encoding and is very cheap to compute.** We perform a comparison among no refinement, one-shot refinement and iterative refinement with synthetic data, where we encode the same function sampled with two different distributions and compute the similarity between the two encodings. Specifically, we generate a random function by

$$f(x) = \frac{1}{2} + \frac{1}{8} \sum_{k=1}^4 a_k \sin(2\pi k x) \tag{14}$$

where $a_k \sim Uniform(0,1)$ are the parameters controlling the generation and $f(x) \in (0,1)$. We construct the encoding of the function by samples from two different sample distributions. The first distribution is computed by $x_i \sim Uniform(0,1)$ and $x_i \leftarrow x_i^2$. The second distribution is computed by $x_i \sim Uniform(0,1)$ and $x_i \leftarrow 1 - x_i^2$. Consequently, the first distribution is left-tailed, and the second distribution is right-tailed. Then we compare the similarity of function encodings generated by no iterative refinement, one-shot refinement, and iterative refinement. Figure 9 shows the comparison, which demonstrates that one-shot refinement is a quite good approximation of the sample invariant function encoding (the similarity increases from $\sim 0.5$ to 0.98 after one-shot refinement). In Appendix I.4, Table 3, we also compare the effectiveness of the three refinement schemes in a synthetic regression problem.

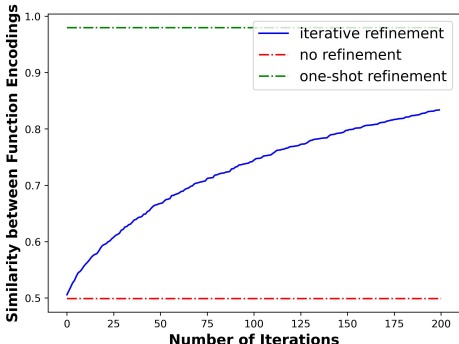

Figure 9: When encoding the same function under two different sample distributions, one-shot refinement can approximate the sample invariant function encoding well. It takes 75/90/1500 ms to encode 5000 samples on a CPU, and 7.5/8.0/250 ms on an NVIDIA Titan-X GPU when performing no refinement/one-shot refinement/200-step iterative refinement.

### I.4 EFFECTIVENESS OF SAMPLE INVARIANCE

In this section, we examine the effectiveness of HDFE's sample invariance property through an synthetic function regression problem. We first generate random functions by equation 14, where $a_k \sim Uniform(0,1)$ are the parameters controlling the generation. The task is to regress the

Table 3: Performance of function parameters regression. PointNet fails when sample distribution varies between training and testing phases, while HDFE is robust to the sample distribution variation.

| | | PointNet | HDFE | | |
| | | | No Refinement | One-Shot Refinement | 200-Step Iter. Ref. |
| --- | --- | --- | --- | --- | --- |
| No Distr. Var. | MSE | 0.0037 | < 0.0005 | < 0.0005 | < 0.0005 |
| | $R^2$ | 0.978 | > 0.9975 | > 0.9975 | > 0.9975 |
| Distr. Var. | MSE | 0.0717 | 0.003 | < 0.0005 | 0.001 |
| | $R^2$ | 0.513 | 0.982 | > 0.9975 | 0.992 |

coefficients $[a_1, a_2, a_3, a_4]$ from the function samples $\{x_i, f(x_i)\}$. Regarding the sample size, the number of function samples is 5000 in the training phase and 2500 in the testing phase. Regarding sample distribution, we consider two different settings:

**Setting 1 (No Sample Distribution Variation)**: The sample distribution is consistent between training phase and testing phase. We let $x_i \sim Uniform(0, 1)$ in both training phase and testing phase.

**Setting 2 (Sample Distribution Variation)**: The sample distribution is different between the training phase and testing phase. Specifically, in the training phase, we let $x_i \sim Uniform(0, 1)$ and $x_i \leftarrow x_i^2$. In the testing phase, we let $x_i \sim Uniform(0, 1)$ and $x_i \leftarrow 1 - x_i^2$. Consequently, the sample distribution in the training phase is left-tailed, while in the testing phase is right-tailed.

We compare our HDFE with PointNet in terms of mean squared error (MSE) and the R-squared ($R^2$) metrics. For HDFE, we compare the performance among no refinement, one-shot refinement (introduced in Appendix I.3), and 200-step iterative refinement. Table 3 shows the comparison.

In Setting 1, when there is no distribution variation, HDFE achieves significantly lower error than PointNet. This is because HDFE is capable of capturing the entire distribution of functions, while PointNet seems to struggle on that.

In Setting 2, when there is distribution variation, PointNet fails miserably, while HDFE, even without iterative refinement, already achieves fairly good estimation, and even better than the PointNet in Setting 1. In addition, the experiment also shows that both the one-shot refinement and the iterative refinement are effective techniques to improve the robustness to distribution variation.

## I.5    INFORMATION LOSS OF HDFE

In this section, we analyze the information loss when encoding continuous objects with HDFE. It is intuitive that a larger encoding dimensionality induces smaller information loss, and encoding a function changing more rapidly induces larger information loss. We attempt to quantify the relation through empirical experiments. We generate random functions and we measure the "function complexity" by the integral of the absolute gradient: $\text{complexity}(f) = \int_0^1 |f'(x)| dx$. Consequently, functions changing more rapidly yield a higher $\text{complexity}(f)$.

We study the relation between the reconstruction mean absolute error (MAE) / R-squared ($R^2$) and the function complexity under different encoding dimensions. Figure 10 reveals the MAE exhibits a linear relation with the input function complexity, while the R-squared seems not to be affected by the function complexity when the dimension is large enough.

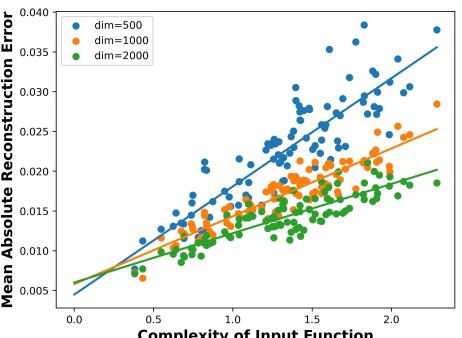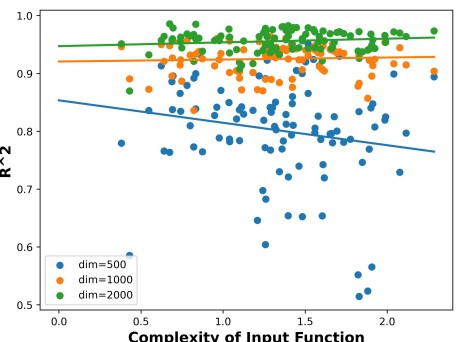

Figure 10: Empirical information loss when encoding functions of different complexities.

### I.6 Low-Rank High-Dimensional Scenarios

We generate random functions by first randomizing $x_k \in \mathbb{R}^d$ and $\alpha_k \in \mathbb{R}$, the random function is constructed by:

$$f(x) = \sum_{k=1}^{n} \alpha_k \cdot K(x, x_k)$$

where $n$ can measure the complexity of the function, and $d$ is the dimension of the function input. The testing samples are generated by $x_k + noise$ for all $k \in [n]$.

In Fig. 11, the reconstruction error increases as $n$ increases, which indicates the encoding quality is negatively correlated to the complexity of the function. However, the reconstruction error does not change as $d$ increases, which indicates that the encoding quality does not depend on the explicit dimension of the function input.

This empirical experiment shows that HDFE has the potential to operate on high-dimensional data, because the encoding quality of HDFE does not depend on the dimension of the function input, but only depends on the complexity of the function.

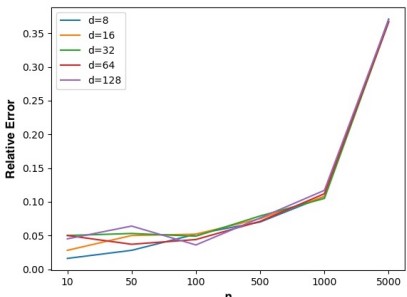

Figure 11: The reconstruction error of HDFE is negatively correlated to the complexity of the function, but does not depend on the dimension of the function input.

## J  Experiment Details

### J.1  PDE Solver

The PDE and the solution are encoded into two embedding vectors with length $N$. A deep complex network is trained to learn the mapping between two vectors. The architecture is a sequence of layers: `[ComplexLinear(N,256), ComplexReLU(), ComplexLinear(256,256), ComplexReLU(), ComplexLinear(256,256), ComplexReLU(), ComplexLinear(256,N)]`. The network is trained with Adam optimizer with a learning rate of 0.001 for 20,000 iterations. The $\alpha$ value in equation 6 is 15, 25, 42, 45 for $N = 4000, 8000, 16000, 24000$ and the $\beta$ value is 2.5.

### J.2  Surface Normal Estimation

The architecture is a sequence of layers: `[ComplexLinear(N,256), ComplexBatchNorm() ComplexReLU(), ComplexLinear(256,256), ComplexBatchNorm(), ComplexReLU(), ComplexLinear(256,128), ComplexBatchNorm(), ComplexReLU()]`. After the sequence of layers, it will produce a 128-dimensional complex vector z. Since we desire a 3-dimensional real vector output (normal vector in $\mathbb{R}^3$), we use two `RealLinear(128,3)` layers `L_real` and `L_imag`. The final output normal vector is `L_real(z.real) + L_imag(z.imag)`. The network is trained with Adam optimizer with learning rate 0.001 for 270 epochs. The $\alpha$ is chosen as 20 and the dimensionality is chosen as 4096.

## J.3 Adding HDFE Module to HSurf-Net

Denote the input local patch as $P$ with shape $(B, N, 3)$, where $B$ is the batch size, $N$ is the number of points in a local patch. HSurf-Net uses their novel space transformation module to extract $n$ keypoints and their $K$ nearest neighbors. The resulting data is denoted as $P\_sub$ with shape $(B, n, K, 3)$. HSurf-Net uses a PointNet to process $P\_sub$, by first lifting the dimension to $(B, n, K, C)$ and then doing a maxpooling to shape the data into $(B, n, C)$. We add our HDFE module here: we use HDFE to shape the data from $(B, n, K, 3)$ to $(B, n, C)$, by first lifting the dimension from $(B, n, K, 3)$ to $(B, n, K, 512)$ using equation 6 and then average the embedding across the neighbors to shape it into $(B, n, 512)$. Then we use a fully-connected layer to map the data into $(B, n, C)$, which becomes the HDFE feature. Then we sum the features generated by HSurf-Net and HDFE into a $(B, n, C)$ matrix and pass to the output layer as HSurf-Net does.

## K Ablation Studies of Surface Normal Estimation

Table 4: Ablation Studies on the PCPNet dataset.

| Dimension | 2048 | | | | | | | 4096 | | | | | | |
|---|---|---|---|---|---|---|---|---|---|---|---|---|---|---|
| Noise Level | None | Low | Med | High | Stripe | Gradient | Average | None | Low | Med | High | Stripe | Gradient | Average |
| $\alpha = 10$ | 8.98 | 10.80 | **17.40** | **22.18** | 10.62 | 9.92 | 13.32 | 9.00 | **10.60** | **17.40** | **22.48** | 10.50 | 9.87 | 13.31 |
| $\alpha = 15$ | 8.98 | 11.02 | 17.73 | 22.72 | 10.85 | 9.46 | 13.46 | 8.29 | 10.77 | 17.46 | 22.63 | 10.17 | 9.10 | 13.07 |
| $\alpha = 20$ | **8.14** | **10.53** | 17.86 | 23.07 | **9.93** | **8.81** | **13.06** | **7.97** | 10.72 | 17.69 | 22.76 | **9.47** | **8.67** | **12.88** |
| $\alpha = 25$ | 8.86 | 11.43 | 17.94 | 22.85 | 10.42 | 9.33 | 13.47 | 8.40 | 11.23 | 17.66 | 22.74 | 9.89 | 8.95 | 13.15 |

Table 5: Ablation Studies on the FamousShape dataset.

| Dimension | 2048 | | | | | | | 4096 | | | | | | |
|---|---|---|---|---|---|---|---|---|---|---|---|---|---|---|
| Noise Level | None | Low | Med | High | Stripe | Gradient | Average | None | Low | Med | High | Stripe | Gradient | Average |
| $\alpha = 10$ | 15.12 | 18.43 | 30.86 | **38.66** | 15.52 | 13.82 | 22.07 | 15.06 | 18.15 | **30.61** | 38.50 | 16.81 | 13.71 | 22.14 |
| $\alpha = 15$ | 14.42 | **17.97** | **30.45** | 38.92 | 15.47 | 13.81 | 21.84 | 13.68 | **17.77** | 31.17 | 38.79 | 14.83 | 13.06 | 21.55 |
| $\alpha = 20$ | **13.37** | 18.43 | 31.40 | 39.03 | **13.92** | **12.54** | **21.45** | **13.04** | 17.99 | 31.23 | **38.57** | **14.01** | **12.13** | **21.16** |
| $\alpha = 25$ | 14.70 | 19.71 | 31.52 | 38.95 | 15.04 | 13.56 | 22.25 | 14.03 | 19.16 | 31.39 | 38.65 | 14.38 | 13.22 | 21.81 |

In general, when the dimensionality is higher, the error is lower. When the receptive field is large ($\alpha$ is small), HDFE performs better when the noise level is high. When the receptive field is small ($\alpha$ is large), HDFE performs better when the noise level is low. This coincides with the analysis in Fig. 2. A large receptive field tends to filter out the perturbations and therefore is more robust to noise. A small receptive field can capture the high-frequency details and therefore is more accurate.

## L Noise Robustness

In this section, we further analyze why HDFE is robust to point perturbations. For visualization purposes, we perform the analysis on 2d data, while the analysis generalizes well to higher dimensions. We randomly sample 1000 points from the unit circle $x^2 + y^2 = 1$ and add Gaussian noises to the samples. Then we encode the points into vectors using HDFE with different receptive fields ($\alpha = 10, 15, 20, 25$) and reconstruct the implicit function. The pseudo-color plot in Fig. 12 visualizes the likelihood that a point lies on the unit circle.

When $\alpha$ is small, the visualization shows that the reconstructions under different noise levels are similar, which tells that the encodings under different noise levels are similar. So it demonstrates a robustness to point perturbations. On the other hand, when $\alpha$ is large, the reconstruction is more refined and captures the high-frequency details, but as a price, it is more sensitive to the point perturbations.

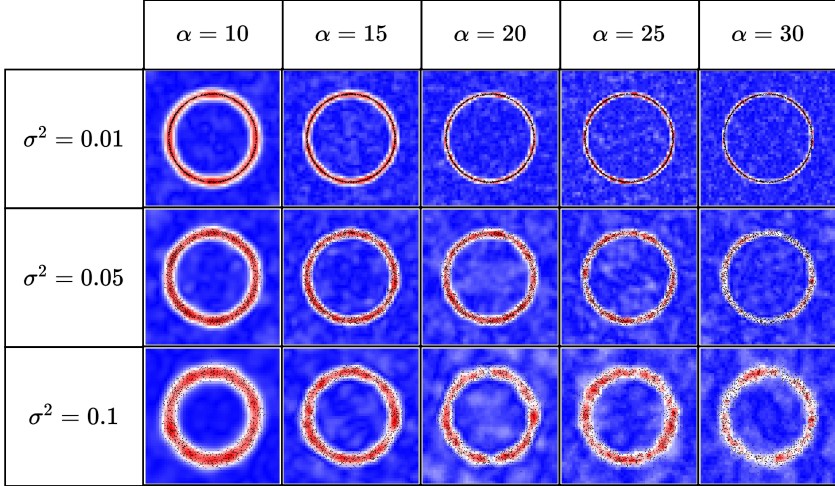

Figure 12: Reconstruction of implicit functions sampled with noisy inputs under different choices of receptive field.

