# OpenReview forum: "Decodable and Sample Invariant Continuous Object Encoder"
_ICLR.cc/2024/Conference — ICLR 2024 poster_

### Official Review · Reviewer_jCpW · 2023-11-01

**Soundness:** 4 excellent
**Presentation:** 4 excellent
**Contribution:** 4 excellent
**Rating:** 8
**Confidence:** 2

**Summary:**

This paper proposes Hyper-Dimensional Function Encoding (HDFE), which does not require training and maps continuous objects for embedding space. The proposed method enables processing continuous objects. Experiments show that the proposed method can be plugged into and improve PointNet-based architectures.

**Strengths:**

1. Encoding continuous signals is an important research topic. The paper is clear and well-organized.
2. The proposed method does not require any-training and can be plugged into existing structures, which makes it easy to apply in practice and could have wide applications.
3. Evaluation is thorough and solid. The method shows advantages over various prior works, across different datasets and settings.

**Weaknesses:**

1. In Table 1, some metrics did not show improvement when comparing to the prior work HSurf-Net.
2. The encoding capacity of the proposed method might be limited.

**Questions:**

Can the proposed method be applied for any-resolution image encoder for complex natural images, e.g. ImageNet? What would the main challanges be for applying the method to the image domain?

---

> ### Author Response · Authors · 2023-11-17
>
> We are sincerely grateful for your positive feedback on our research topic, contributions, and the organization of the paper. Your insightful questions have not only deepened our understanding but also guided us towards future research directions. We are pleased to address your concerns as follows:
>
> > Weakness 1: In Table 1, some metrics did not show improvement when comparing to the prior work HSurf-Net.
>
> We acknowledge the reviewer's observation regarding Table 1's metrics in comparison to HSurf-Net. We understand the importance of this concern and have provided a detailed analysis in the global response section. This analysis explores the potential reasons behind these findings and their broader implications.
>
> > Weakness 2: The encoding capacity of the proposed method might be limited.
>
> We recognize the concern about HDFE's limited encoding capacity for extensive input spaces, a point we also note in our conclusions. To address this, we propose partitioning the input space into disjoint subspaces and concatenating their respective encodings. While our current focus has been on low-dimensional functions, where capacity is less of an issue, expanding HDFE’s capacity for broader applications presents a significant future research direction.
>
> > Question 1: Can the proposed method be applied for any-resolution image encoder for complex natural images, e.g. ImageNet? What would the main challanges be for applying the method to the image domain?
>
> The application of HDFE to complex natural image tasks, such as those involving ImageNet, introduces specific challenges, particularly its lack of translation invariance. This means the HDFE encoding for an image and its shifted version would differ significantly, making it less suitable for tasks requiring translation invariance like classification and detection. Nonetheless, HDFE may be well-suited for image-related tasks that do not necessitate translation invariance, such as image-to-image translation or image super-resolution. This area presents exciting avenues for future exploration, and we are grateful for your question, which has stimulated further consideration of HDFE's potential in image processing.

---

### Official Review · Reviewer_421T · 2023-11-01

**Soundness:** 3 good
**Presentation:** 2 fair
**Contribution:** 2 fair
**Rating:** 6
**Confidence:** 4

**Summary:**

This submission propose a module, namely Hyper-Dimension Function Encoding (HDFE), to map a continuous object (data sample) into a fixed-dimension vector without any training.
The author asserts that the proposed approach possesses four key characteristics: (1) sample distribution invariance (2) sample size invariance (3) explicit representation (4) decodability.
To obtain the fixed-length vector representation, the input data for HDFE must adhere to Lipschitz continuity and will be transformed into a high-dimensional space, where a weighted average will be computed.

**Strengths:**

1. The manuscript demonstrates excellent organization, a well-defined research problem, clear logic, and skillful writing.

2. The topic holds significant importance: a method that can map data samples with varying distributions and sizes to fixed-length sequences may be highly appealing for pre-training models that utilize cross-domain data.

3. The theory effectively connects with the experiment: the utilization of a weighted average operation has the potential to reduce noise effectively.

**Weaknesses:**

1. It is necessary to provide a clear definition of "implicit representation" and "explicit representation" in the manuscript. Some reviewers may intuitively refer to the fixed-length vector representation of a data sample as "implicit representation" since it may not be human-friendly. However, in this manuscript, the fixed-length vector representation is referred to as the "explicit representation."
2. The proposed method (HDFE) relies on the assumption that the input data follows Lipschitz continuity. While the reviewer agrees that point cloud data intuitively follows Lipschitz continuity, it would be beneficial for the manuscript to include an analysis of the types of input data that adhere to Lipschitz continuity.

3. As a module that doesn't require any training, it is important to provide detailed guidance on selecting hyperparameters. This includes guidance on choosing the size of the fixed-length vector representation (denoted as $N$) and determining the hyperparameters $\alpha$ and $\beta$ in Equation 5, which are influenced by the receptive field $\epsilon_0$ and the Lipschitz continuous constant $c$.

4. 2The selection of weights ($w_i$ in Equation 1), hyperparameters ($\alpha$ and $\beta$), and the mapping functions $E_X$ and $E_Y$ are highly dependent on the dataset. This means that if the task or input data changes, all these variables need to be carefully decided and tested.

5. There is a small concern regarding the experimental results on the PCPNet dataset. The proposed HDFE method is demonstrated to outperform the PCPNet model (the baseline in 2018) simply by replacing PointNet with HDFE. However, it is only comparable to the current state-of-the-art (SOTA) method, outperforming it in four out of twelve metrics, albeit with a slight drop in average performance. It would be valuable to provide insights or explanations for these observations and discuss any potential limitations or implications of the results.

**Questions:**

1. In line 7 of page 2, why is the representation learned by PointNet (Qi et al., 2017a) not easily decodable? For instance, in their original paper (https://arxiv.org/pdf/1612.00593.pdf) in Figure 2, it seems possible to set m=3 and obtain normalized point clouds. Additionally, other works like [1] may also be able to 'decode' the input from the vector representation. Is there any difference between this manuscript and those works?

2. When experimenting with batches, should the model visit all data samples to decide hyperparameters? According to Equation 3, the decoding step should visit all $\bm Y$.

3. By curious: why a high-dimensional input does not affect the size of the fixed-length vector representation $N$. Could the author provide further explanation, possibly an extension of the paragraph on 'Scale to high-dimensional input'?

4. in Section 2.3, the manuscript only shows the picking of $E_X$ and $E_Y$ when the function output y is scalar. Are there more cases that can be considered?

5. Although HDFE is a deterministic function, is there any empirical result available to estimate the information loss from the raw input to the fixed-length vector representation?



[1] Learning Representations and Generative Models for 3D Point Clouds. PMLR

---

> ### Author Response · Authors · 2023-11-17
> **Answering Questions raised by Reviewer 421T**
>
> > Question 1: Why is the representation learned by PointNet (Qi et al., 2017a) not easily decodable?
>
> Thank you for your insightful question. We claim that PointNet's representations are not easily decodable based on our experiments where PointNet, as a function encoder, struggled to minimize reconstruction loss and failed to yield reasonable reconstructions. Detailed findings are presented in **Appendix B: PointNet as Function Encoder**.
>
> While previous works like [1] may encode implicit functions (e.g., point cloud inputs) effectively, our analysis suggests **they lack the capability for decodable representation of explicit functions**, as evidenced in Appendix B. Additionally, these methods **may not produce encoding invariant to sample distribution changes**. For instance, if trained on uniformly sampled point clouds and tested on varied sampling distributions, their performance could deteriorate, as discussed in Appendix I.4.
>
> We are thankful for this question, which has enhanced the completeness of our manuscript, and we are open to any further inquiries regarding our experiments or arguments.
>
> [1] Learning Representations and Generative Models for 3D Point Clouds. PMLR
>
> > Question 2: When experimenting with batches, should the model visit all data samples to decide hyperparameters?
>
> **The hyperparameters $\alpha$, $\beta$, and the mappings $E_X$, $E_Y$ are predetermined and remain constant during both training and testing.** The selection process, particularly for $\alpha$, involves identifying hyperparameters that minimize the function reconstruction loss. Once selected, these hyperparameters are fixed, eliminating the need for adjustments across batches. It's important to note that for meaningful comparisons of function encodings, they must be generated using consistent hyperparameters.
>
> > Question 3: why a high-dimensional input does not affect the size of the fixed-length vector representation. Could the author provide further explanation, possibly an extension of the paragraph on 'Scale to high-dimensional input'?
>
> Our conclusion regarding the negligible impact of high-dimensional input on the size of the fixed-length vector representation is based on empirical findings. Detailed in Appendix I.6, our experiments assess the influence of input dimension and function complexity on reconstruction error. These studies reveal that **while function complexity significantly affects the reconstruction error, the input dimension appears to have a minimal impact**. This leads us to conjecture that HDFE can effectively encode high-dimensional data without incurring information loss due to increased data dimension.
>
> We appreciate the reviewer's interest in this topic, aligning with our discussion in the "Scale to high-dimensional input" section. Our intention is to highlight a promising research avenue and provide a foundational understanding of its viability.
>
> > Question 4: in Section 2.3, the manuscript only shows the picking of and when the function output y is scalar. Are there more cases that can be considered?
>
> Thank you for prompting further consideration of HDFE's application scope. To address functions with vector outputs, HDFE can be easily adapted by reshaping the $\Psi$ matrix in Equation 6. For cases where function inputs or outputs cannot be directly vectorized using Equation 6, we propose pretraining the input/output mappings $E_X$, $E_Y$ to align with the properties outlined in Section 2.1. This approach would enable HDFE to handle non-vector functions by employing customized mappings, as the principles for designing these mappings are thoroughly discussed in Section 2.1.
>
> > Question 5: Although HDFE is a deterministic function, is there any empirical result available to estimate the information loss from the raw input to the fixed-length vector representation?
>
> To address the query about HDFE's information loss, we've included **Appendix I.5: Information Loss of HDFE** in our manuscript. Our empirical analysis reveals a linear relationship between information loss (quantified by reconstruction error) and function complexity, where complexity is measured by the integral of the absolute gradient. Interestingly, we observe that the reconstruction $R^2$ value appears uncorrelated with complexity in cases of sufficiently high input dimension. We trust this analysis will be insightful for the reviewer. We are grateful for this question, which has contributed to the manuscript's comprehensiveness.

---

> ### Author Response · Authors · 2023-11-17
> **Addressing Weaknesses raised by Reviewer 421T**
>
> We are grateful for the careful attention and time you have dedicated to reviewing our manuscript. Your constructive feedback is invaluable, and we have made concerted efforts to address each of your concerns, incorporating additional experiments and clarifications into the revised manuscript.
>
> > Weakness 1: It is necessary to provide a clear definition of "implicit representation" and "explicit representation" in the manuscript.
>
> Thank you for highlighting the need for clearer definitions of "implicit representation" and "explicit representation" in our manuscript. We acknowledge the potential confusion among readers regarding these terms. In response to your suggestion, we have revised the definition of explicit representation in the Introduction's second paragraph: "Explicit representation refers to frameworks that generate outputs with fixed dimensions, such as fixed-length vectors." To enhance clarity, we have also included a brief comparison between explicit and implicit representations, elucidating their distinct roles and applications in our research context. This modification should provide readers with a more comprehensive understanding and eliminate ambiguities related to these key terms.
>
> > Weakness 2: The proposed method (HDFE) relies on the assumption that the input data follows Lipschitz continuity. It would be beneficial for the manuscript to include an analysis of the types of input data that adhere to Lipschitz continuity.
>
> Thank you for highlighting this aspect. In response, we have included **Appendix D: Suitable Input Types for HDFE**, detailing various input types that conform to Lipschitz continuity. This addition aims to clarify the characteristics of appropriate inputs, offering both theoretical understanding and practical guidance for researchers applying our method.
>
> > Weakness 3: As a module that doesn't require any training, it is important to provide detailed guidance on selecting hyperparameters.
>
> We concur that providing guidance for hyperparameter selection is crucial for HDFE's usability. We contend, however, that hyperparameter tuning should not pose a significant barrier to users. HDFE's simplicity is characterized by minimal hyperparameter requirements. Specifically, for dimensionality, higher values invariably yield superior representations, obviating the need for intricate tuning.
> The primary hyperparameter, the receptive field, is governed by the $\alpha$ parameter in Equation 6. Typically, $\alpha$ ranges from 10 to 30 for normalized inputs between $(0,1)$, simplifying the selection process.
> Moreover, we plan to release HDFE's source code along with comprehensive documentation, including hyperparameter guidance. We also envision incorporating automated mechanisms for receptive field optimization based on user-specific datasets, further reducing the tuning burden.
>
> > Weakness 4: The selection of weights ($w_i$ in Equation 1), hyperparameters ($\alpha$ and $\beta$), and the mapping functions and are highly dependent on the dataset. This means that if the task or input data changes, all these variables need to be carefully decided and tested.
>
> We appreciate the opportunity to clarify potential misunderstandings:
>
> **Task Change vs. Input Functions**: If the task changes (e.g., from classification to regression) but the input functions remain constant, the hyperparameters need not be altered. Only the downstream neural network requires modification.
>
> **Weights $w_i$**: These are not hyperparameters but are determined through iterative refinement during input function encoding, thereby eliminating the need for user tuning.
>
> **Hyperparameter $\beta$**: This remains constant as long as the function's range is normalized between 0 and 1. For functions normalized within $(0,1)$, we set $\beta$ to 2.5, ensuring the gradient of $\langle E_Y(y_1), E_Y(y_2)\rangle$ stays positive, as detailed in the manuscript.
>
> **Changes in Input Functions**: The reselection of the $\alpha$ value (in Equation 4) is only necessary if there's a significant change in the input functions (e.g., shifting from 1D to 2D functions) or in the Lipschitz constant.
>
> Please note that the hyperparameters $\alpha$, $\beta$, and the mappings $E_X$, $E_Y$ are fixed across both training and testing phases to ensure comparability of function encodings.
>
> > Weakness 5: There is a small concern regarding the experimental results on the PCPNet dataset. It is only comparable to the current state-of-the-art (SOTA) method, outperforming it in four out of twelve metrics, albeit with a slight drop in average performance.
>
> We acknowledge the reviewer's concern regarding the experimental results with the PCPNet dataset. To address this, we have conducted a thorough analysis of the HDFE's performance compared to both the baseline and current state-of-the-art methods. For a comprehensive understanding, we invite the reviewer to refer to our detailed analysis presented in the global response section.

---

### Official Review · Reviewer_oyrE · 2023-11-09

**Soundness:** 4 excellent
**Presentation:** 3 good
**Contribution:** 3 good
**Rating:** 8
**Confidence:** 3

**Summary:**

This paper proposes Hyper-Dimensional Function Encoding (HDFE), which encodes a continuous object (eg, functions) into a fixed-size explicit vector representation without requiring training. While it maintains the benefits of vector function architecture (VFA), satisfying sample invariance and decodability, it relaxes the strict assumption on the function form in VFA into Lipschitz functions by introducing a novel iterative refinement process. While HDFE serves as a general interface for a continuous object encoder without training, substituting HDFE for domain-specific algorithms in experiments on mesh-grid data and sparse data shows comparable performance.
The main contributions of the papers are:

(1) The authors propose a novel function encoding method that satisfies key properties of VFA while relaxing the strict assumption on function space to  Lipschitz continuity.

(2) Theoretical foundations and empirical analysis support the validity of HDFE on key properties.

(3) Experimental results confirm that replacing domain-specific algorithms with HDFE maintains competitive performance and robustness to the noise perturbation.

**Strengths:**

- The paper is well written and easy to follow.

- The formulation of the decodable encoder and iterative refinement process for sample invariance seems interesting and convincing. Also, theoretical analysis on each component is clear and supports the claims.

- Despite the general and straightforward formulation, empirical results demonstrate the effectiveness of HDFE.

**Weaknesses:**

Method

- One concern is the computational cost of HDFE induced by the iterative refinement process. In order to employ function representation for the downstream tasks, computational costs of HDFE is important. it may hinder application to large-scale tasks.

Experiment

- Overall, it’s convincing that HDFE is a reasonable and general interface for processing continuous objects, supported by the experiments. However, it’s less convincing why we should use HDFE instead of other domain-specific encoding methods. The authors claim that sample invariance is a crucial property for the machine learning tasks throughout the paper, but it lacks the supporting experiment revealing HDFE’s efficacy in those scenarios (i.e., sample distributions are different in training and test dataset). It would make the paper stronger if it presents the experiments with scenarios having disparate sample distributions between training and test datasets and compares the performance of HDFE compared to the baselines.

- In the experiment section, it lacks the analysis why HDFE is more beneficial than the counterparts (e.g., PointNet) in terms of the performance. It would improve the understanding of HDFE if analysis on which component leads to the performance gap even when the noise is absent is provided.

**Questions:**

- How long does the HDFE take compared to the baselines (eg, pointNet in Experiment 3.2)? Is the iterative refinement process applicable to a large number of samples? How long does it take for convergence in the process?

- In the formulation on decoding, (i.e., equation between eq. (2) and eq.(3)), can you please clarify on why orthogonality property ensures that $E_X(x_i) ⊘ E_X(x_0) $ will produce a vector orthogonal to $E_X(x_0)$ when the distance between two samples is large? Also what does the noise mean? Does it mean that it’s near zero so that it is a negligible component?

- In the formulation on decoding, (i.e., equation between eq. (2) and eq.(3)), it seems it misses $w_i$.

- For an unbinding operation, element-wise division of complex vectors is used. But I don't think this operation is commutative, which violates the assumption. Can you please clarify on this?

- In experiment 3.1, how does the function prediction error is measured? Is it measured in embedding space? And the paper states that “when decoding is not required, our approach achieves lower error than FNO”, but how can we compare to FNO, which directly predicts the solution?

- While the authors claim that HDFE is robust to point perturbation, the experiments on  [PCPNet - PointNet + HDFE] in Table 1 shows that the performance boost becomes much less as the noise level increases. Can you please elaborate on this?

- [Possible Typo] In the last sentence in section 2.1, “appendix F.1” should be “appendix E.1”.

---

> ### Author Response · Authors · 2023-11-17
> **Addressing Weaknesses raised by Reviewer oyrE**
>
> We are grateful for the time and attention you have devoted to carefully reading our manuscript. We are eager to address your constructive concerns and have incorporated additional experiments and clarifications in the revised manuscript. We intent to make our replies elegant and concise for easy understanding, and we encourage Reviewer oyrE to refer to the corresponding sections in the updated manuscript for detailed experimental insights.
>
> > Weakness 1: One concern is the computational cost of HDFE induced by the iterative refinement process.
>
> We acknowledge that the need for iterative refinement, which entails additional computational effort, presents a significant challenge for large-scale tasks. However, in fact, **the refinement process can be efficiently executed in a single iteration, yielding a highly accurate estimation of the sample invariant function encoding**.
>
> **Details:** The motivation of the iterative refinement is to balance the weights between dense and sparse samples. Such motivation can also be achieved by another cheaper one-shot refinement. After obtaining the initial function encoding by averaging the sample encoding, we compute the similarity between this initial encoding and all the sample encodings. The similarity can serve as a rough estimation of the sample density at a particular point. Therefore, if we were to balance the weights between dense and sparse samples, we can simply recompute the weights by the inverse estimated density. It turns out that **this one-shot refinement can achieve similarity of > 0.95 when sample distribution varies.** **The computational cost is only 8.0 ms on an NVIDIA Titan-X GPU.**
>
> We kindly refer Reviewer oyrE to **Appendix I.3: Practical Consideration of Iterative Refinement** in the updated manuscript for the detailed procedure of one-shot refinement and experiments demonstrating its effectiveness. We highly appreciate your very constructive question that induces us to complete the manuscript.
>
> ---
>
> > Weakness 2: The authors claim that sample invariance is a crucial property for the machine learning tasks throughout the paper, but it lacks the supporting experiment revealing HDFE’s efficacy in those scenarios.
>
> We heartly agree that experiments supporting HDFE's efficacy on sample invariance are important. **We will show HDFE is significantly more robust to distribution variation than PointNet through two experiments**: 1. point cloud normal estimation; 2. a synthetic function parameters regression problem.
>
> 1. **Point Cloud Normal Estimation**
>
>    In our updated experiments, **HDFE significantly outperforms the PointNet counterpart when sample variance exists.** Specifically, HDFE achieves 4.79/7.37/0.52/0.46 lower errors than its PointNet counterpart in the setting of Density-Gradient (i.e. the density of the input point cloud varies) in the two baseline comparisons and two benchmark datasets. Through the comparison with the baseline, it is convincing that the significant improvement is attribute to HDFE's sample invariance property and the augmentation of the HDFE encoding.
>
>    We highly appreciate your very constructive feedback and we have added a paragraph "**HDFE promotes stronger robustness to point density variance**." in the **Section 3.2: Unoriented Surface Normal Estimation** to make the analysis and discussion explicit to readers.
>
> 2. **Synthetic Function Parameters Regression**.
>
>    We generate random functions parametrized by four coefficients and the task is to predict the four coefficients using the function samples. We artificially control the sample distribution to be varied or not varied across the training and testing phase. It turns out that **PointNet's prediction $R^2$ decreases from 0.978 to 0.513** when switching from no variation to variation, while **HDFE maintains its $R^2$ at > 0.9975** in both settings. We kindly refer the Reviewer oyrE to **Appendix I.4: Effectiveness of Sample Invariance** in the updated manuscript for the experiment details and analysis.
>
> These experiments demonstrate HDFE's superior performance in scenarios with sample variance, thereby validating our claims about its efficacy in machine learning tasks. We highly appreciate your very constructive question that induces us to complete the manuscript.
>
> ---
>
> > Weakness 3: In the experiment section, it lacks the analysis why HDFE is more beneficial than the counterparts (e.g., PointNet) in terms of the performance. It would improve the understanding of HDFE if analysis on which component leads to the performance gap even when the noise is absent is provided.
>
> We agree on your concern and we have investigated into the cause. We kindly refer the reviewer to the posted global response, where we explain our analysis in details.

---

> ### Author Response · Authors · 2023-11-17
> **Answering Questions from Reviewer oyrE, Part 2/2**
>
> > Question 6: While the authors claim that HDFE is robust to point perturbation, the experiments on [PCPNet - PointNet + HDFE] in Table 1 shows that the performance boost becomes much less as the noise level increases. Can you please elaborate on this?
>
> We believe it is caused by the trade-off between accuracy and robustness to noises. In Table 1, we present the result obtained by $\alpha=20$, which minimizes the average error. However, this parameter, compared with other selections of $\alpha$, does not obtain lowest error at all noise levels. Kindly refer to Appendix K, where we find a larger receptive field makes HDFE more robust to noise, while a smaller receptive field makes HDFE more accurate in clean inputs. We can increase HDFE's robustness to noise by choosing a large receptive field, but we will lose some accuracy when the input is clean.
>
> > Question 7: [Possible Typo]
>
> Thanks for your catch! We have corrected the typo.

---

> ### Author Response · Authors · 2023-11-17
> **Answering Questions raised by Reviewer oyrE, Part 1/2**
>
> > Question 1: How long does the HDFE take compared to the baselines (eg, pointNet in Experiment 3.2)? Is the iterative refinement process applicable to a large number of samples? How long does it take for convergence in the process?
>
> While the details are in the response to Weakness 1, we would like to mention that HDFE is much faster than PointNet. For an input of shape $5000\times 3$, assuming the complexity of multiplying an $m\times n$ matrix with an $n\times p$ matrix is $mnp$, then HDFE only has has complexity $5000\times3\times1024\approx1.5e7$. The computational cost of the one-shot refinement is negligible here. But PointNet has complexity $5000\times3\times64+5000\times64\times128+5000\times128\times1024\approx 70e7$.
>
> > Question 2: In the formulation on decoding, (i.e., equation between eq. (2) and eq.(3)), can you please clarify on why orthogonality property ensures that will produce a vector orthogonal to when the distance between two samples is large? Also what does the noise mean? Does it mean that it’s near zero so that it is a negligible component?
>
> We recognize that the noise argument may not be straight-forward from the orthogonal property. For better rigor, clarification and readability, we replace the orthogonal property as similarity-preserving property, without affecting the correctness of HDFE:
>
> ​	similarity preserving: ⟨x ⊗ y, x ⊗ z⟩ = ⟨y, z⟩.
>
> When $d_X(x_0, x_i)$ is large, $\langle E_X(x_0)\otimes E_Y(f(x_0)), E_X(x_i)\otimes E_Y(f(x_i)) \rangle\approx 0$. Since the unbinding operation is similarity preserving, we have $\langle E_Y(f(x_0)), E_X(x_i)\otimes E_Y(f(x_i)) \oslash E_X(x_0)\rangle\approx 0$. Therefore, $\mathbf{F}\oslash E_X(x_0)$ can be decomposed into $E_Y(f(x_0))$ and the sum of vectors that are orthogonal to it. Consequently, when we search for the $y$ to maximize the $\langle \mathbf{F}\oslash E_X(x_0), E_Y(y) \rangle$, those orthogonal vectors will not bias the optimization.
>
> Note that the Fractional Power Encoding does satisfy the similarity preserving property, so the entire HDFE formulation is still correct. We have rewritten the **decoding** section based on the new definition and put the argument in **Appendix E: Derivation of Decoding**. We highly appreciate your constructive question for helping us improve the manuscript.
>
> > Question 3: In the formulation on decoding, it seems it misses $w_i$.
>
> Thanks for your catch! We have corrected the typo.
>
> > Question 4: For an unbinding operation, element-wise division of complex vectors is used. But I don't think this operation is commutative, which violates the assumption. Can you please clarify on this?
>
> This is a very sharp catch! We admit it is an overlook by us when we draft the manuscript. You are correct that the element-wise division of complex vectors is not commutative. Actually the commutative property is not required when deducing the decoding formulation, so the HDFE formulation is still correct. We have revised the definition of our binding and unbinding operations to address this (and Question 2) and ensure the logical consistency of our methodology.
>
> > Question 5: In experiment 3.1, how does the function prediction error is measured?
>
> The function prediction error is computed by the mean absolute error between the predicted function and the ground-truth function, as mentioned in the Section 3.1 -> Dataset.
>
> > Question 5: And the paper states that “when decoding is not required, our approach achieves lower error than FNO”, but how can we compare to FNO, which directly predicts the solution?
>
> We first encode the ground-truth solution into a vector and then decode the vector to reconstruct the solution. From this, we obtain a reconstruction loss $error_1$, which is visualized by the shallow red bars (Reconstruction Error) in Figure 3. Then we decode the predicted function vector to reconstruct the solution. From this, we obtain the error between the predicted solution and the ground-truth solution, denoted by $error_2$, which is visualized by the stack of Reconstruction Error and Function Prediction Error.
>
> Our argument is, $error_2$ consists of two components. The first component arises when reconstructing the function from the encoding, which is measured by $error_1$. The second component arises when predicting the function encoding, which cannot be measured directly. But since $error_2$ only consists of two components, we can estimate the error arised from the second component by $error_2-error_1$, which is visualized as the Function Prediction Error in Figure 3.

---

### Author Response · Authors · 2023-11-17
**Performance Gap between [HSurf-Net] and [HSurf-Net + HDFE]**

One concern raised by all reviewers is **the performance gap between [HSurf-Net] and [HSurf-Net + HDFE] in several settings.** We heartly agree on the concern and we investigate into the cause. We found that the performance gap is mainly caused by the **inappropriate fusion of HSurf-Net and HDFE**. By addressing the issue, **we are able to eliminate the gap and make [HSurf-Net + HDFE] better than [HSurf-Net] in all settings.** Reviewers can refer to the updated Table 1 for the improvement. Here are the details:

In our original experiment, we *concatenated* the HDFE features with the HSurf-Net features and fed the concatenated features to the output layer, as introduced in Appendix I (old manuscript). It turns out that such concatenation fusion causes some overhead and degredates the performance. In our updated experiment, we replace the *concatenation* with *addition*, which yields a significantly better performance.

Specifically, denoting the input local patch as $P$ with shape $(B, N, 3)$, where $B$ is the batch size, $N$ is the number of points in a local patch. HSurf-Net uses their space transformation module to select $n$ keypoints and their $K$ nearest neighbors. The resulting data is denoted as  $P$_ $sub$ with shape $(B, n, K, 3)$. HSurf-Net uses a PointNet to process $P$_ $sub$, by first lifting the dimension to $(B, n, K, C)$ and then doing a maxpooling to shape the data into $(B, n, C)$, which becomes the HSurf-Net feature. Then we compute the HDFE feature by lifting the dimension of   $P$_ $sub$, into $(B, n, K, 512)$ using Equation 6 and then average the embedding across the neighbors to shape the data into $(B, n, 512)$. Then we use a fully-connected layer to map the data into $(B, n, C)$, which becomes the HDFE feature.  After obtaining the HSurf-Net feature and the HDFE feature, instead of concatenating them into a tensor of shape $(B, n, 2C)$, we sum the feature to form a tensor of shape $(B, n, C)$. Then we feed the fused feature into an output layer as HSurf-Net does. The details have been placed at **Appendix J.3** in the updated manuscript.

In addition to changing the feature fusion, **we also discover that augmenting encoding generated by HDFE can improve the performance significantly.** Specically, when encoding local patches using Equation 4, we add noises to the computed weight $w_i$ during training. It turns out the augmentation can boost the performance significantly, especially when there is density variation. In the updated Table 1, we compare the effect of augmentation in both baselines. We also add a paragraph "**HDFE promotes stronger robustness to point density variance.**" to discuss the benefit of HDFE's sample invariance property.

**[Change of Manuscript]**

* We remove Figure 4 and Figure 5 since they do not contribute much to the analysis and we are allocating the space for the discussion of robustness to point density variance due to the page limit.

* We rewrite Section 3.2, especially the discussion paragraphs, and update Table 1 to accommodate our new findings.
* In Table 1, since we do not compare with any methods from Jet to SHS-Net, we remove the corresponding rows to make the comparison more focused.

**[Supplementary Comments]**

Reviewers may notice in our updated experiments, the error in the high-noise setting is higher than that in our old experiments. We believe it is caused by the trade-off between the accuracy and the robustness. In the old experiment, we use a large receptive field ($\alpha=10$), while in the updated experiment, we use a small receptive field ($\alpha=20$). We notice that a large receptive field is more robust to noise but less accurate when noise is absent. In contrast, a small receptive is more accurate when noise is absent but less robust to noise. Such trade-off can be seen at Appendix K and Appendix L (revised manuscript).

---

### Author Response · Authors · 2023-11-17
**Manuscript Modification Summary - Writing**

* **Avoid Ambiguity of "Explicit Representation"**. We thank the suggestion by Reviewer 421T to avoid potential confusion and ambiguity caused by "explicit representation". In the introduction, 2nd paragraph,

  > (3) *Explicit representation*: the framework can encode the continuous objects into explicit representations.

  is changed into

  > (3) *Explicit representation*: the framework generates outputs with fixed dimensions, such as fixed-length vectors.

* **Clarify the definition of unbinding operation**. We appreciate that Reviewer oyrE points out a mis-definition of the unbinding operation about the commutative property. This is an overlook by us when drafting the manuscript: the unbinding operation is not required to be commutative. We have corrected the definition and derivation of decoding in Section 2.1.

  > 4. invertible: there exists an associative inverse operator, called *unbinding* ⊘, such that x ⊘ (x ⊗ y) = (x⊘x)⊗y = y. The unbinding operator is also commutative, distributive and orthogonal.

  is changed into

  > 4. invertible: there exists an associative, distributive, orthogonal operator that undoes the binding, called *unbinding* ⊘, satisfying (x ⊗ y) ⊘ z = (x ⊘ z) ⊗ y and (x ⊗ y) ⊘ x = y.

  And we add an explanation of the binding and unbinding operations to help readers better understand:

  > The binding and unbinding operations can be analogous to multiplication and division, where the difference is that binding and unbinding operate on vectors ...

  In the derivation, all the $E_X(x_0)\oslash E_X(x_i)\otimes E_Y(f(x_i))$ is changed into $ E_X(x_i)\otimes E_Y(f(x_i))\oslash E_X(x_0)$ to accomodate the corrected definition.

* **Explain the formulation of noise**. We appreciate the question from Reviewer oyrE to clarify why $E_X(x_i)\otimes E_Y(f(x_i))\oslash E_X(x_0)$ produces a noise vector with respect to $E_Y(f(x_0))$. We admit that the reasoning in the manuscript is not straight-forward and may cause some confusion.

  We add **Appendix E: Derivation of Decoding** to provide further details on the decoding. In addition, for better clarification and easier understanding, we change the orthogonal property required by the unbinding operation into similarity preserving property. All the derivation in the manuscript is still correct under this modifed definition.

  > 3. orthogonal: ⟨x⊗y,x⟩≈0, ⟨x⊗y,y⟩≈0 when ⟨x,y⟩ ≈ 0.

  is changed into

  > 3. similarity preserving: ⟨x ⊗ y, x ⊗ z⟩ = ⟨y, z⟩.

* **Rewrite contributions due to the updated experiments.** To accomodate our updated experiments, we modify the abstract and introduction to convey the message of "incorporating HDFE into the PointNet-based SOTA architecture leads to average error decreases of 2.5% and 1.7%.".

* **Remove Figure 4 and Figure 5 to leave spaces for the discussion paragraphs,** due to the page limit.

* **Other miscellaneous changes of rephrasing, adding reference to appendices.**

---

### Author Response · Authors · 2023-11-17
**Manuscript Modification Summary - Updated Experiments**

* **Update experiments and add discussions at Section 3.2 Unoriented Surface Normal Estimation**. We appreciate the concerns from all the reviewers about the performance gap between [HSurf-Net] and [HSurf-Net + HDFE]. We update the experiments, rewrite the discussions and add a new paragraph "*HDFE promotes stronger robustness to point density variance*" to discuss the benefit of HDFE's sample invariance property. Now, [HSurf-Net + HDFE] is better than [HSurf-Net] in all the settings.
* **Add Appendix B: PointNet as Function Encoder**. We appreciate the question from Reviewer 421T about why PointNet cannot produce a decodable representation. In this section, we attempt to train a PointNet function encoder by minizing the reconstruction loss. It turns out that PointNet fails in this task and does not generate reasonable reconstruction. Therefore, we infer that without substantial modification, PointNet is unlikely to acquire the necessary capability for function encoding.
* **Add Appendix D: Suitable Input Types for HDFE**. We appreciate the question from Reviewer 421T which requests to include suitable types of input data that adhere to Lipschitz continuity. In this section, we provide three real-world data types that we believe are suitable as inputs for HDFE, namely point cloud local geometry, meteorological measurements, and sparse time series data. This is not an exhaustive enumeration of all suitable inputs, but we hope to provide a conceptual understanding of the characteristics of a suitable input.
* **Add Appendix I.3: Practical Consideration for Iterative Refinement**. We appreciate the question from Reviewer oyrE about the computational cost of iterative refinement. In this section, we demonstrate that the refinement can be done in one shot to achieve good approximation (similarity > 0.95) of the sample invariant function encoding. The computational cost of achieving an approximate sample invariant representation from 5000 samples is only 8.0 ms on a GPU.
* **Add Appendix I.4: Effectiveness of Sample Invariance**. We appreciate the question from Reviewer oyrE about the effectiveness of HDFE's sample invariance property. In this section, we perform function parameters prediction using synthetic data, where we artificially create sampling discrepancy between training and testing set. The experiment shows that PointNet fails miserably when sampling discrepancy exists, while HDFE does not experience any performance degradation.
* **Add Appendix I.5: Information Loss of HDFE**. We appreciate the question from Reviewer 421T about the information loss of HDFE. In this section, we provide some empirical insight, where we study how the reconstruction loss increases when the complexity of the function increases. The reconstruction error exhibits a linear relation between the absolute gradient integration $\int_0^1 |f'(x)| dx$, while the reconstruction $R^2$ seems uncorrelated to the function complexity.

---

### Author Response · Authors · 2023-11-17
**Summary of Global Replies**

We sincerely thank all the reviewers for their careful reading and very insightful questions. We are grateful that all reviewers place positive feedback on the sample invariance research question, the elegance and effectiveness of our proposed HDFE, and the experiment design. We are more than pleased to address the concerns raised by the reviewers, which significantly improves the manuscript.

In the global thread replies, we will first address the concerns raised by all the three reviewers. Then we summarize the modification of the manuscript and the new experiments requested by reviewers. We heartly welcome any further discussion with the reviewers.

We are currently in the process of drafting detailed responses for each individual reviewer. In our forthcoming responses, we will specifically address the unique questions raised by each reviewer, acknowledging the valuable insights they have provided.

---

### Meta-Review · Area_Chair_j1Ui · 2023-12-11

**Metareview:**

The paper investigates an approach to encode continuous objects into a fixed-dimensional vector. To this end, the authors identified four desiderata for such an encoder, namely (1) sample distribution invariance (2) sample size invariance (3) explicit representation (4) decodability.

The paper received generally positive reviews from three reviewers. The primary concerns raised by the reviewers were about (1) marginal improvement over some baseline (HSurf-Net),  (2) justifications on the Lipschitz continuity assumption, and (3) clarification of some expressions.  The authors adequately addressed some of the concerns, especially regarding the experiments, and the reviewers maintained their initial recommendation. After reading the paper, reviews, and rebuttal, the AC agrees with the reviewers’ decision and recommends acceptance. The authors should include the additional results and clarifications presented in the rebuttal to the camera-ready version.

**Justification For Why Not Higher Score:**

Despite the generality in principle, the method is evaluated with one synthetic and real-world data, respectively.

**Justification For Why Not Lower Score:**

N/A

---

### Decision · Program_Chairs · 2024-01-16

Accept (poster)